# Early dysfunction and progressive degeneration of the subthalamic nucleus in mouse models of Huntington's disease

Jeremy F Atherton, Eileen L McIver, Matthew RM Mullen, David L Wokosin, D James Surmeier, Mark D Bevan*

Department of Physiology, Feinberg School of Medicine, Northwestern University, Chicago, United States

**Abstract** The subthalamic nucleus (STN) is an element of cortico-basal ganglia-thalamo-cortical circuitry critical for action suppression. In Huntington's disease (HD) action suppression is impaired, resembling the effects of STN lesioning or inactivation. To explore this potential linkage, the STN was studied in BAC transgenic and Q175 knock-in mouse models of HD. At <2 and 6 months of age autonomous STN activity was impaired due to activation of $K_{ATP}$ channels. STN neurons exhibited prolonged NMDA receptor-mediated synaptic currents, caused by a deficit in glutamate uptake, and elevated mitochondrial oxidant stress, which was ameliorated by NMDA receptor antagonism. STN activity was rescued by NMDA receptor antagonism or the break down of hydrogen peroxide. At 12 months of age approximately 30% of STN neurons had been lost, as in HD. Together, these data argue that dysfunction within the STN is an early feature of HD that may contribute to its expression and course.

## Introduction

The basal ganglia are a network of subcortical brain nuclei that are critical for action selection and central to the expression of several psychomotor disorders (*Albin et al., 1989*; *Wichmann and DeLong, 1996*). Information flow from the cortex to the output nuclei of the basal ganglia occurs via three major pathways. The so-called direct pathway through the striatum promotes movement and 'rewarding' behavior through inhibition of GABAergic basal ganglia output (*Chevalier and Deniau, 1990*; *Kravitz et al., 2010*; *Kravitz and Kreitzer, 2012*). In contrast, the indirect pathway via the striatum, external globus pallidus and subthalamic nucleus (STN) and the hyperdirect pathway through the STN suppress the same processes through elevation of basal ganglia output (*Maurice et al., 1999*; *Tachibana et al., 2008*; *Kravitz et al., 2010*; *Kravitz and Kreitzer, 2012*). Indeed, interruption of the indirect and hyperdirect pathways through lesion or inactivation of the STN is associated with elevated/involuntary movement, impulsivity and psychiatric disturbances such as hypomania and hyper-sexuality (*Crossman et al., 1988*; *Hamada and DeLong, 1992*; *Baunez and Robbins, 1997*; *Bickel et al., 2010*; *Jahanshahi et al., 2015*).

Huntington's disease (HD) is an autosomal dominant, neurodegenerative disorder caused by an expansion of CAG repeats in the gene (*HTT*) encoding huntingtin (HTT), a protein involved in vesicle dynamics and intracellular transport (*Huntington's Disease Collaborative Research Group, 1993*; *Saudou and Humbert, 2016*). Early symptoms of HD include involuntary movement, compulsive behavior, paranoia, irritability and aggression (*Anderson and Marder, 2001*; *Kirkwood et al., 2001*). These symptoms have traditionally been linked to cortico-striatal degeneration, however a role for the STN is suggested by their similarity to those caused by STN inactivation or lesion. The hypoactivity of the STN in HD models in vivo (*Callahan and Abercrombie, 2015a*, *2015b*) and the

*For correspondence: m-bevan@northwestern.edu

Competing interests: The authors declare that no competing interests exist.

susceptibility of the STN to degeneration in HD (*Lange et al., 1976*; *Guo et al., 2012*) are also consistent with STN dysfunction.

Several mouse models of HD have been generated, which vary by length and species origin of *HTT/Htt*, CAG repeat length, and method of genome insertion. For example, one line expresses full-length human HTT with 97 mixed CAA-CAG repeats in a bacterial artificial chromosome (BAC; *Gray et al., 2008*), whereas Q175 knock-in (KI) mice have an inserted chimeric human/mouse exon one with a human polyproline region and ~188 CAG repeats in the native *Htt* (*Menalled et al., 2012*).

Increased mitochondrial oxidant stress exacerbated by abnormal NMDAR-mediated transmission and signaling has been reported in HD and its models (*Fan and Raymond, 2007*; *Song et al., 2011*; *Johri et al., 2013*; *Parsons and Raymond, 2014*; *Martin et al., 2015*). Several reports suggest that glutamate uptake is impaired due to reduced expression of the glutamate transporter EAAT2 (GLT-1) and/or GLT-1 dysfunction (*Arzberger et al., 1997*; *Liévens et al., 2001*; *Behrens et al., 2002*; *Miller et al., 2008*; *Bradford et al., 2009*; *Faideau et al., 2010*; *Huang et al., 2010*; *Menalled et al., 2012*; *Dvorzhak et al., 2016*; *Jiang et al., 2016*). However, others have found no evidence for deficient glutamate uptake (*Parsons et al., 2016*), suggesting that abnormal NMDAR-mediated transmission is caused by increased expression of extrasynaptic receptors and/or aberrant coupling to signaling pathways (e.g., *Parsons and Raymond, 2014*). The sensitivity of mitochondria to anomalous NMDAR signaling is likely to be further compounded by their intrinsically compromised status in HD (*Song et al., 2011*; *Johri et al., 2013*; *Martin et al., 2015*).

Although HD models exhibit pathogenic processes seen in HD, they do not exhibit similar levels and spatiotemporal patterns of cortico-striatal neurodegeneration. Striatal neuronal loss in aggressive *Htt* fragment models such as R6/2 mice does occur but only close to death (*Stack et al., 2005*), whereas full-length models exhibit minimal loss (*Gray et al., 2008*; *Smith et al., 2014*). Despite the loss and hypoactivity of STN neurons in HD and the similarity of HD symptoms to those arising from STN lesion or inactivation, the role of the STN in HD remains poorly understood. We hypothesized that the abnormal activity and progressive loss of STN neurons in HD may reflect abnormalities within the STN itself. This hypothesis was addressed in the BAC and Q175 KI HD models using a combination of cellular and synaptic electrophysiology, optogenetic interrogation, two-photon imaging and stereological cell counting.

## Results

Data are reported as *median [interquartile range]*. Unpaired and paired statistical comparisons were made with non-parametric Mann-Whitney U and Wilcoxon Signed-Rank tests, respectively. Fisher's exact test was used for categorical data. $p < 0.05$ was considered statistically significant; where multiple comparisons were performed this p-value was adjusted using the Holm-Bonferroni method (adjusted p-values are denoted $p_h$; *Holm, 1979*). Box plots show median (central line), interquartile range (box) and 10–90% range (whiskers).

### The autonomous activity of STN neurons is disrupted in the BACHD model

STN neurons exhibit intrinsic, autonomous firing, which contributes to their role as a driving force of neuronal activity in the basal ganglia (*Bevan and Wilson, 1999*; *Beurrier et al., 2000*; *Do and Bean, 2003*). To determine whether this property is compromised in HD mice, the autonomous activity of STN neurons in ex vivo brain slices prepared from BACHD and wild type littermate (WT) mice were compared using non-invasive, loose-seal, cell-attached patch clamp recordings. 5–7 months old, symptomatic and 1–2 months old, presymptomatic mice were studied (*Gray et al., 2008*). Recordings focused on the lateral two-thirds of the STN, which receives input from the motor cortex (*Kita and Kita, 2012*; *Chu et al., 2015*). At 5–7 months, 124/128 (97%) WT neurons exhibited autonomous activity compared to 110/126 (87%) BACHD neurons ($p = 0.0049$; *Figure 1A,B*). The frequency (WT: 7.9 [5.2–12.6] Hz; n = 128; BACHD: 6.3 [1.4–10.2] Hz; n = 126; $p = 0.0001$) and regularity (WT CV: 0.27 [0.14–0.47]; n = 124; BACHD CV: 0.36 [0.20–0.80]; n = 110; $p = 0.0012$) of firing were reduced in BACHD neurons (*Figure 1A,B*). The distribution of firing frequency of WT neurons appears unimodal with a mode at ~6–8 Hz (*Figure 1C*), whereas the distribution of BACHD neurons is relatively bimodal with modes at ~0–2 Hz and ~8–10 Hz (Kolmogorov–Smirnov test, $p = 0.0002$;

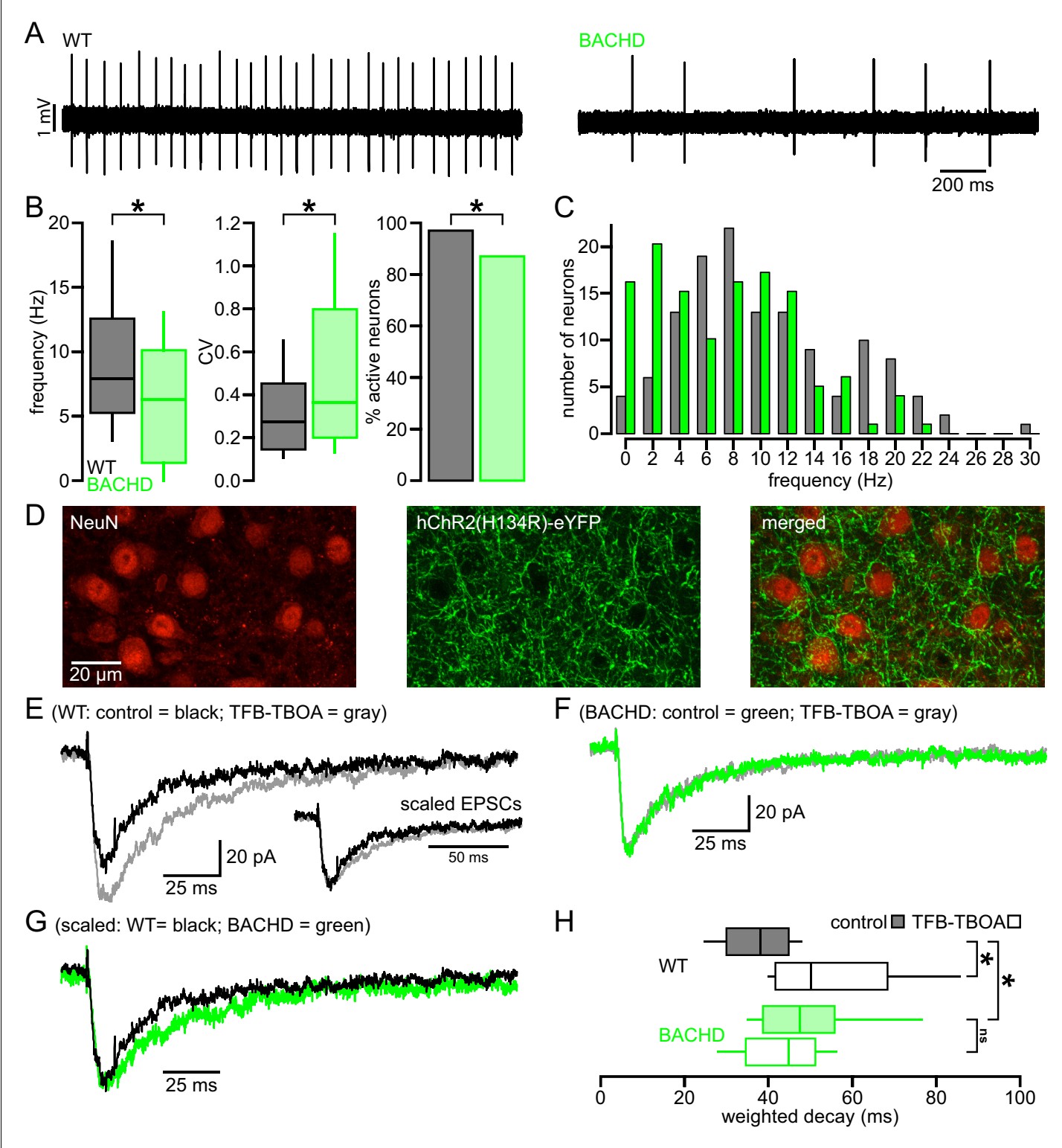

**Figure 1.** Abnormal intrinsic and synaptic properties of STN neurons in BACHD mice. (**A**) Representative examples of autonomous STN activity recorded in the loose-seal, cell-attached configuration. The firing of the neuron from a WT mouse was of a higher frequency and regularity than the phenotypic neuron from a BACHD mouse. (**B**) Population data showing (left to right) that the frequency and regularity of firing, and the proportion of active neurons in BACHD mice were reduced relative to WT mice. (**C**) Histogram showing the distribution of autonomous firing frequencies of neurons in WT (gray) and BACHD (green) mice. (**D**) Confocal micrographs showing NeuN expressing STN neurons (red) and hChR2(H134R)-eYFP expressing cortico-STN axon terminals (green) in the STN. (**E**) Examples of optogenetically stimulated NMDAR EPSCs from a WT STN neuron before (black) and

*Figure 1 continued on next page*

*Figure 1 continued*

after (gray) inhibition of astrocytic glutamate uptake with 100 nM TFB-TBOA. Inset, the same EPSCs scaled to the same amplitude. (F) Examples of optogenetically stimulated NMDAR EPSCs from a BACHD STN neuron before (green) and after (gray) inhibition of astrocytic glutamate uptake with 100 nM TFB-TBOA. (G) WT (black, same as in E) and BACHD (green, same as in F) optogenetically stimulated NMDAR EPSCs overlaid and scaled to the same amplitude. (H) Boxplots of amplitude weighted decay show slowed decay kinetics of NMDAR EPSCs in BACHD STN neurons compared to WT, and that TFB-TBOA increased weighted decay in WT but not BACHD mice. *p < 0.05. ns, not significant. Data for panels **B–C** provided in *Figure 1—source data 1*; data for panel H provided in *Figure 1—source data 2*.

The following source data is available for figure 1:

**Source data 1.** Autonomous firing frequency and CV for BACHD and WT STN neurons in *Figure 1B–C*.
**Source data 2.** Amplitude weighted decay of NMDAR-mediated EPSCs in *Figure 1H*.

*Figure 1C*). This distribution suggests that BACHD neurons consist of a phenotypic population with compromised autonomous firing, and a non-phenotypic population with relatively normal autonomous firing. At 1–2 months 136/145 (94%) WT STN neurons were autonomously active versus 120/143 (84%) BACHD STN neurons (p = 0.0086). The frequency (WT: 9.8 [6.3–14.8] Hz; n = 145; BACHD: 7.1 [1.8–11.3] Hz; n = 143; p < 0.0001) and regularity (WT CV: 0.17 [0.13–0.26]; n = 136; BACHD CV: 0.23 [0.14–0.76]; n = 120; p = 0.0016) of firing were also reduced in BACHD neurons. Together, these data demonstrate that the autonomous activity of STN neurons in BACHD mice is impaired at both early presymptomatic and later symptomatic ages.

## NMDAR-mediated EPSCs are prolonged in BACHD STN neurons

As described above, the majority of studies report that astrocytic glutamate uptake is diminished in the striatum in HD and its models. To test whether the STN of BACHD mice exhibits a similar deficit, EPSCs arising from the optogenetic stimulation of motor cortical inputs to the STN (as described by *Chu et al., 2015*) were compared in WT and BACHD mice before and after inhibition of GLT-1 and GLAST with 100 nM TFB-TBOA. STN neurons were recorded in ex vivo brain slices in the whole-cell voltage-clamp configuration using a cesium-based, QX-314-containing internal solution to maximize voltage control. Neurons were held at −40 mV and recorded in the presence of low (0.1 mM) external $Mg^{2+}$ and the AMPAR antagonist DNQX (20 µM) to maximize and pharmacologically isolate the evoked NMDAR-mediated excitatory postsynaptic current (EPSC); analysis was performed on average EPSCs from 5 trials with 30 s recovery between trials (*Figure 1D–H*). The amplitude weighted decay time constant of the NMDAR EPSC was moderately but significantly prolonged in BACHD compared to WT neurons (WT: 38.1 [30.0–44.8] ms; n = 12; BACHD: 47.6 [38.7–55.9] ms; n = 11; p = 0.0455; *Figure 1E–H*). Subsequent application of TFB-TBOA increased the decay time constant of the NMDAR EPSC in STN neurons derived from WT (WT control: 39.0 [35.2–44.0] ms; WT TFB-TBOA: 50.2 [41.7–68.4] ms; n = 9; p = 0.0039; *Figure 1E,H*) but had no effect in BACHD neurons (BACHD control: 47.9 [38.9–59.4] ms; BACHD TFB-TBOA: 44.9 [34.7–52.2] ms; n = 10; p = 0.3223; *Figure 1F,H*). In control conditions, the amplitudes of EPSCs recorded from WT and BACHD neurons were similar (WT: 50.1 [34.7–61.0] pA; n = 12; BACHD: 45.6 [22.1–78.3] pA; n = 11; $p_h$ = 0.7399; *Figure 2A*) and there was no correlation between EPSC amplitude and the decay time constant in either group (WT: $r^2$ = 0.16; n = 12; $p_h$ = 0.5871; BACHD: $r^2$ = 0.10; n = 12; $p_h$ = 0.6686; *Figure 2B*). In order to increase spillover of glutamate from synaptic release sites, cortico-STN inputs were optogenetically stimulated 5 times at 50 Hz and the resulting compound NMDAR-mediated EPSC was compared in WT and BACHD STN neurons. Interestingly, the decay of compound NMDAR EPSCs under control conditions or following inhibition of glutamate uptake were not different in WT and BACHD mice (WT control: 79.0 [62.6–102.0] ms; n = 6; BACHD control: 65.2 [44.7–111.5] ms; n = 6; p = 0.4848; WT TFB-TBOA: 125.4 [106.8–146.6] ms; n = 6; BACHD TFB-TBOA: 108.3 [94.5–143.1] ms; n = 6; p = 0.6991; *Figure 2C–E*). Together, these data demonstrate that individual, but not compound, NMDAR-mediated cortico-STN synaptic EPSCs are prolonged in the BACHD model.

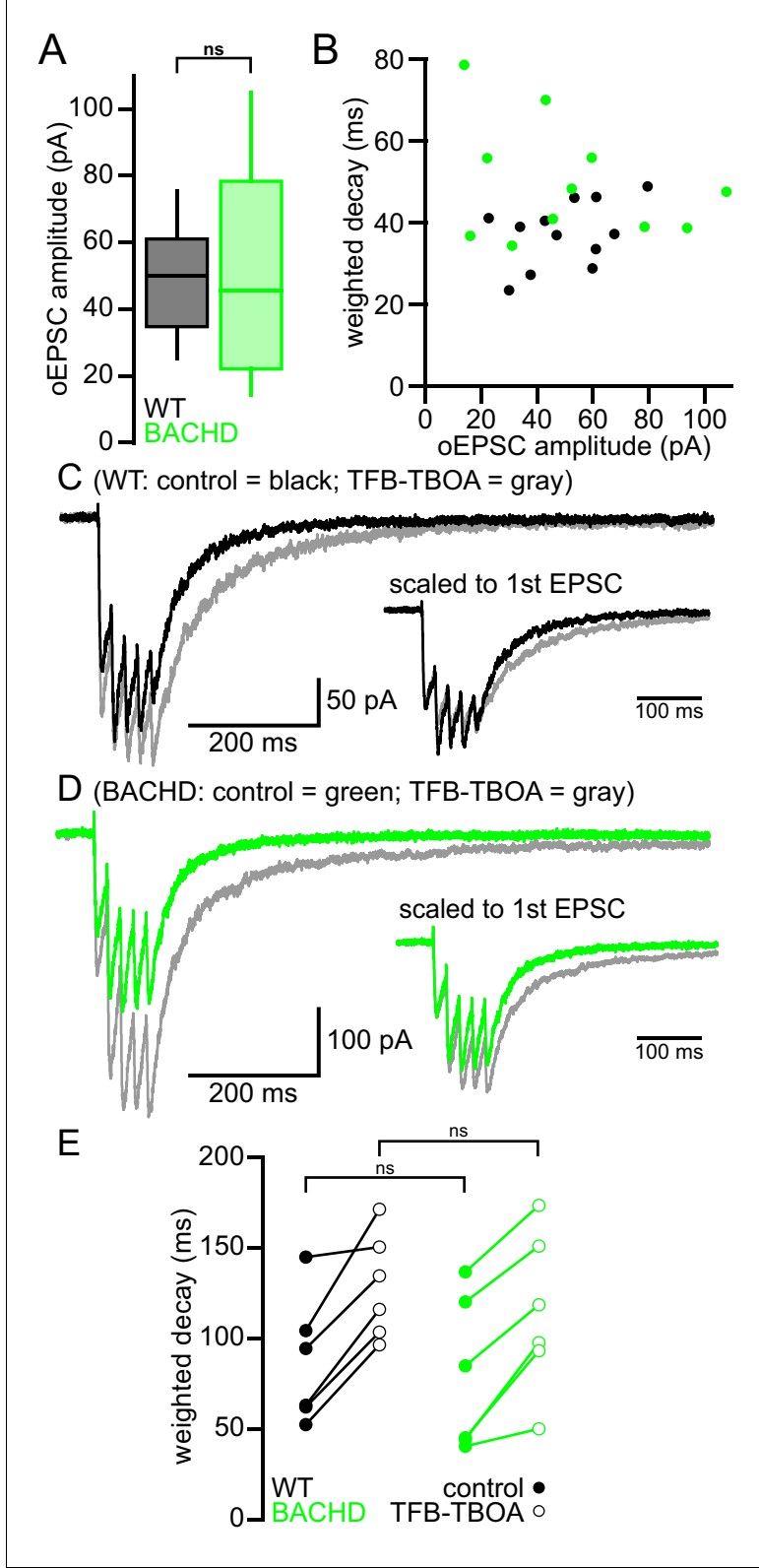

**Figure 2.** Cortico-STN EPSCs in WT and BACHD mice. (**A**) Boxplot showing the distribution of single optogenetically stimulated cortico-STN NMDAR EPSC amplitudes in WT and BACHD mice. (**B**) Scatterplot showing NMDAR EPSC amplitude vs amplitude weighted decay time. There was no correlation between NMDAR EPSC amplitude and decay time in WT or BACHD mice. (**C–D**) Example of NMDAR EPSCs generated by 5 × 50 Hz

*Figure 2 continued on next page*

*Figure 2 continued*

optogenetic stimulation from a WT STN neuron (**C**) before (black) and after (gray) inhibition of astrocytic glutamate uptake with 100 nM TFB-TBOA and a BACHD STN neuron (**D**) before (green) and after (gray) TFB-TBOA application. (**E**) Line segment plots of amplitude weighted decay of compound NMDAR EPSCs before and following TFB-TBOA. The decays of compound NMDAR ESPCs were similar in WT and BACHD before TFB-TBOA application. In addition, inhibition of astrocytic glutamate uptake prolonged the decay of compound NMDAR ESPCs in all neurons tested. ns, not significant. Data for panels **A–B** provided in *Figure 2—source data 1*; data for panel **E** provided in *Figure 2—source data 2*.

The following source data is available for figure 2:

**Source data 1.** Amplitude and amplitude weighted decay of NMDAR-mediated EPSCs in *Figure 2A–B*.

**Source data 2.** Amplitude weighted decay of compound NMDAR-mediated EPSCs in *Figure 2E*.

## Blockade of NMDARs rescues the autonomous activity of BACHD STN neurons

To test whether disrupted autonomous firing in BACHD is linked to NMDAR activation, brain slices from BACHD mice were incubated in control media or media containing the NMDAR antagonist D-AP5 (50 µM) for 3–5 hr prior to loose-seal, cell-attached recordings from STN neurons (*Figure 3*). D-AP5 treatment rescued autonomous firing in slices derived from 5–7 month old BACHD mice compared to untreated control slices (*Figure 3A,B*). The proportion of autonomously active neurons was greater in D-AP5 pre-treated slices (untreated: 18/30 (60%); D-AP5 treated: 29/30 (97%); $p = 0.0011$). The frequency (untreated: 1.0 [0.0–7.6] Hz; n = 30; D-AP5 treated: 13.2 [7.9–17.4] Hz; n = 30; $p < 0.0001$) and regularity (untreated CV: 0.43 [0.24–1.21]; n = 18; D-AP5 treated: CV: 0.13 [0.09–0.20]; n = 29; $p < 0.0001$) of autonomous firing were also greater in D-AP5 treated slices. In slices derived from 1–2 month old BACHD mice autonomous firing was also more prevalent in D-AP5 treated slices than in untreated slices (untreated: 10/30 (33%); D-AP5-treated: 27/30 neurons (90%); $p < 0.0001$) and the frequency of firing overall was greater (untreated: 0.0 [0.0–1.3] Hz; n = 30; D-AP5 treated: 8.7 [4.4–14.5] Hz; n = 30; $p < 0.0001$; *Figure 3B*). The regularity of autonomous firing was however not rescued (untreated CV: 0.61 [0.27–0.81]; n = 10; D-AP5-treated CV: 0.25 [0.14–0.69]; n = 27; $p = 0.1368$; *Figure 3B*). In slices from WT mice the rate (untreated: 10.4 [6.1–14.2] Hz; n = 27; D-AP5 treated: 5.6 [1.7–15.4] Hz; n = 27; $p = 0.1683$; *Figure 3B*), regularity (untreated: 0.18 [0.12–0.27]; n = 24; D-AP5 treated: 0.22 [0.12–0.46]; n = 22; $p = 0.4785$; *Figure 3C*) and incidence of firing (untreated: 24/27 (89%); D-AP5 treated 22/27 (81%); $p = 0.7040$; *Figure 3D*) were unaltered by D-AP5 treatment. Thus, prolonged blockade of NMDARs rescued autonomous firing in BACHD STN neurons but had no effect on autonomous activity in WT STN neurons. Together, these data demonstrate that NMDAR activation contributes to the disruption of autonomous activity in BACHD STN neurons.

## The mitochondria of BACHD STN neurons are subject to elevated oxidant stress

NMDAR activation can elevate mitochondrial oxidant stress (*Dugan et al., 1995*; *Moncada and Bolaños, 2006*; *Brennan et al., 2009*; *Nakamura and Lipton, 2011*). To test whether STN neurons from BACHD mice exhibit increased mitochondrial oxidant stress, a mitochondria-targeted redox probe MTS-roGFP (*Hanson et al., 2004*) was virally expressed in 5–7-month-old BACHD mice and WT littermates (*Figure 4A*). 1–2 weeks later MTS-roGFP was imaged in brain slices under two-photon excitation with 890 nm light. Oxidant stress was estimated from the fluorescence of MTS-roGFP in individual neurons under baseline conditions relative to the fluorescence of MTS-roGFP under conditions of full reduction and oxidation in the presence of 2 mM dithiothreitol and 200 µM aldrithiol-4, respectively (*Sanchez-Padilla et al., 2014*). STN neurons were selected for analysis based on their appearance under two-photon, Dodt contrast imaging and were distinguishable from STN glial cells by their relatively large diameter (*Figure 4A*). STN neurons are reliably recorded when this selection strategy is employed to guide patch clamp recording (*Atherton et al., 2008, 2010*). MTS-roGFP imaging revealed that relative oxidant stress in BACHD STN neurons was elevated compared

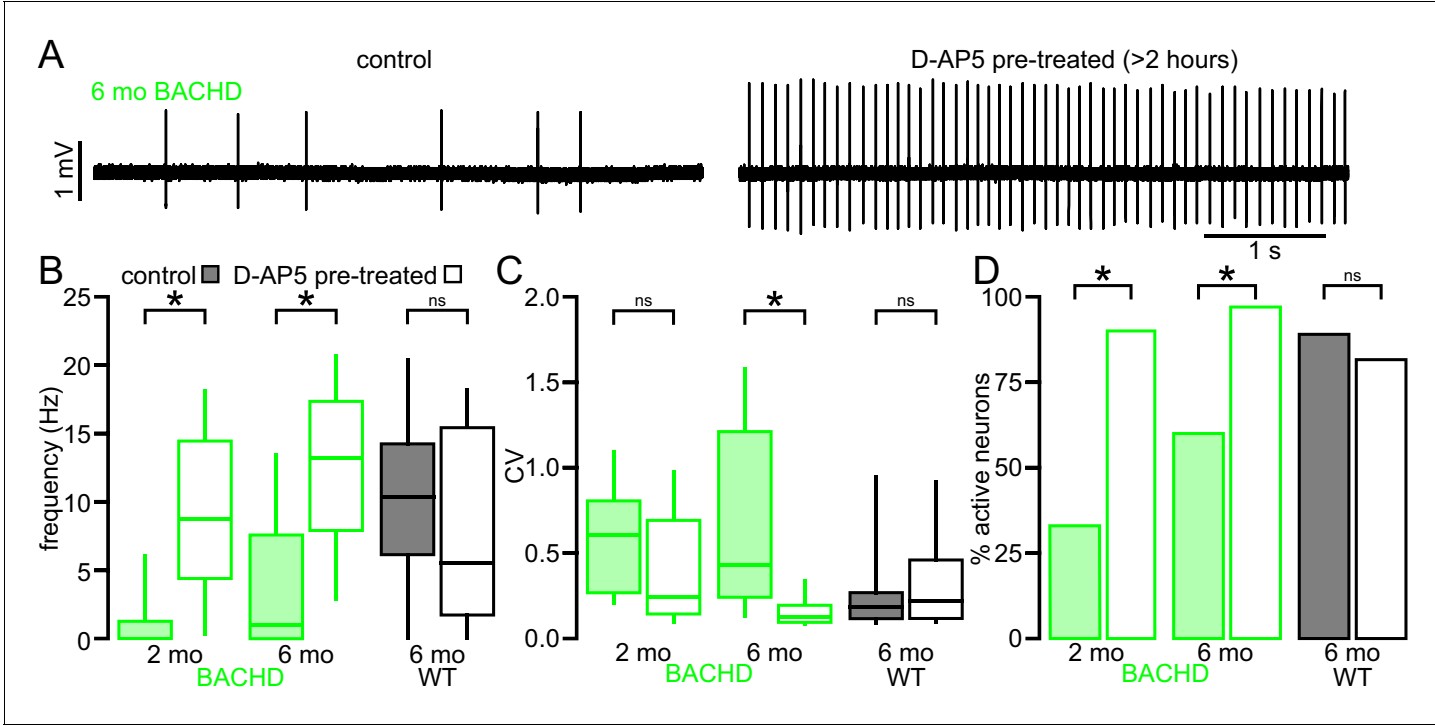

**Figure 3.** The impaired autonomous activity of STN neurons in BACHD mice is rescued by antagonism of NMDARs. (**A**) Examples of loose-seal cell-attached recordings of 6-month-old BACHD STN neurons from an untreated slice (left) and a slice that was treated with 50 μM D-AP5 for 3–5 hr prior to recording. (**B**) Population data showing elevated autonomous firing in STN neurons from D-AP5 pre-treated slices from 2-month-old and 6-month-old BACHD mice but not in WT mice. (**C**) Population data showing increased firing regularity in STN neurons from D-AP5 pre-treated slices from 6-month-old BACHD mice but not 2-month-old BACHD mice or WT mice. (**D**) Population data showing a higher proportion of active neurons in STN neurons from D-AP5 pre-treated slices from 2-month-old and 6-month-old BACHD mice but not in WT mice. *p < 0.05. ns, not significant. Data for panels B–C provided in *Figure 3—source data 1*.

The following source data is available for figure 3:

**Source data 1.** Autonomous firing frequency and CV for BACHD control and D-AP5 pretreated STN neurons in *Figure 3B–C*.

to WT (WT: 0.35 [0.18–0.42]; n = 23; BACHD: 0.40 [0.37–0.63]; n = 24; p = 0.0332; *Figure 4B*). In a separate experiment (performed 15 months later) to test whether NMDAR activation ex vivo contributed to elevated mitochondrial oxidant stress, brain slices from a different cohort of BACHD mice were treated for >3 hr in 50 μM D-AP5 prior to imaging. MTS-roGFP imaging confirmed that D-AP5-treated slices exhibited lower oxidant stress compared to untreated slices from the same mice (untreated: 0.39 [0.35–0.48]; n = 13; D-AP5-treated: 0.32 [0.24–0.42]; n = 17; p = 0.0445; *Figure 4C*). Thus, STN neurons from BACHD mice exhibit elevated mitochondrial oxidant stress, which can be reduced by antagonism of NMDARs.

## Impaired autonomous activity of STN neurons in BACHD mice is due to increased activation of $K_{ATP}$ channels

NMDAR receptor-generated mitochondrial oxidant stress in BACHD may lead to the activation of $K_{ATP}$ channels, which act as metabolic sensors and homeostatic regulators of excitability in multiple cell types (*Nichols, 2006*). STN neurons abundantly express all the molecular components of $K_{ATP}$ channels including the Kir6.2 pore-forming subunit of the $K_{ATP}$ channel (*Thomzig et al., 2005*) and the SUR1, SUR2A and SUR2B regulatory subunits (*Karschin et al., 1997*; *Zhou et al., 2012*). To determine whether $K_{ATP}$ channels are responsible for impaired firing in BACHD mice, the effects of the $K_{ATP}$ channel inhibitor glibenclamide (100 nM) on WT and phenotypic BACHD autonomous firing ex vivo were compared. Glibenclamide application increased both the rate (BACHD control: 5.7 [1.4–9.1] Hz; BACHD glibenclamide: 11.3 [9.2–13.3] Hz; n = 15; p = 0.0003) and regularity (BACHD

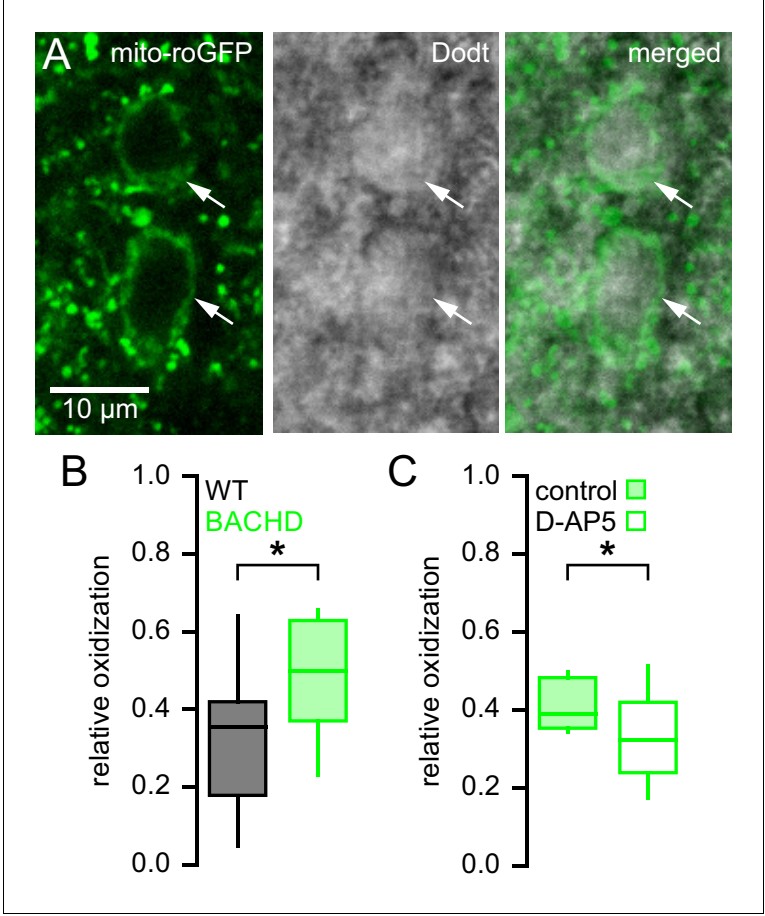

**Figure 4.** Mitochondrial oxidant stress is elevated in STN neurons from BACHD mice. (A) Two-photon (left), Dodt contrast (center) and combined (right) images showing expression of MTS-roGFP (green) in STN neurons of a BACHD mouse. (B) Population data illustrating greater relative oxidation of mitochondria in BACHD STN neurons compared to WT STN neurons. (C) Population data showing that >3 hr pre-treatment with the NMDA receptor antagonist D-AP5 (50 μM) lowered the relative oxidation of mitochondria in BACHD STN neurons. Arrows indicate MTS-roGFP expressing STN neurons; *p < 0.05. Data for panel B provided in *Figure 4—source data 1*; data for panel C provided in *Figure 4—source data 2*.

The following source data is available for figure 4:

**Source data 1.** The relative mitochondrial oxidant stress of WT and BACHD STN neurons in *Figure 4B*.

**Source data 2.** The relative mitochondrial oxidant stress of control and D-AP5 pre-treated BACHD STN neurons in *Figure 4C*.

control CV: 0.41 [0.16–1.43]; BACHD glibenclamide CV: 0.16 [0.12–0.32]; n = 14; p = 0.0001) of autonomous firing in 5–7-month-old BACHD STN neurons (*Figure 5A,B*). In contrast $K_{ATP}$ channel inhibition had no effect on the firing of WT neurons (WT control frequency: 13.2 [8.4–19.6] Hz; WT glibenclamide frequency: 15.7 [9.7–18.4] Hz; n = 8; p = 0.4609; WT control CV: 0.18 [0.11–0.21]; WT glibenclamide CV: 0.14 [0.11–0.19]; n = 8; p = 0.1094).

To further examine the effects of $K_{ATP}$ channels on autonomous firing, whole-cell current clamp recordings were obtained from 5–7-month-old BACHD mice and WT littermates (*Figure 5C,D*). Consistent with the hyperpolarizing and shunting effects of $K_{ATP}$ channels, the interspike voltage trajectory was shallower in BACHD neurons compared to WT (WT: 413.8 [317.1–705.0] mV/s; n = 7; BACHD: 125.3 [59.2–298.0] mV/s; n = 7; $p_h$ = 0.0210). In addition, the rate, regularity and interspike voltage trajectory of autonomous firing of BACHD STN neurons recorded in this configuration

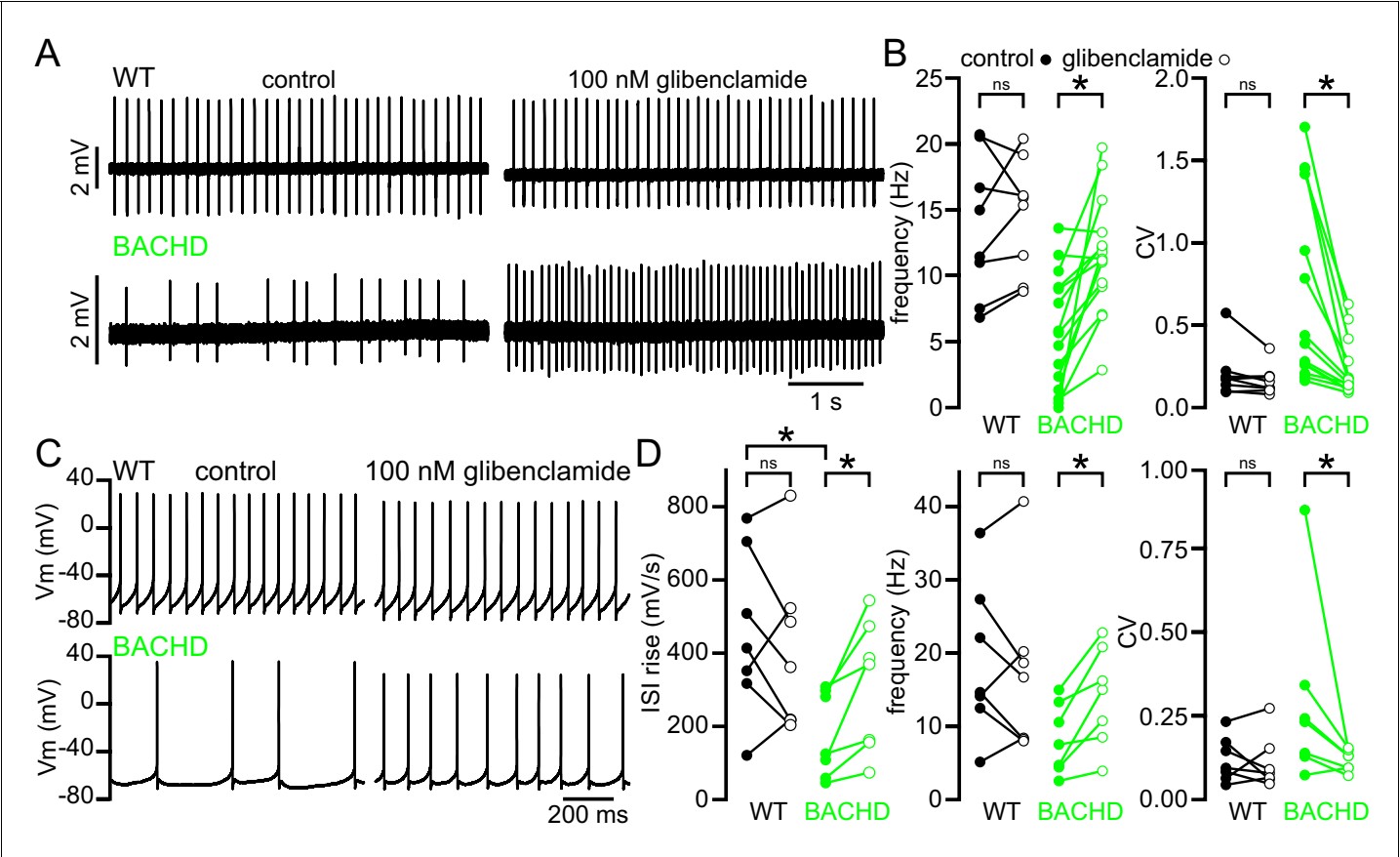

**Figure 5.** The abnormal autonomous activity of STN neurons in BACHD mice is rescued by inhibition of $K_{ATP}$ channels. (A) Examples of loose-seal cell-attached recordings from STN neurons before (left) and after (right) inhibition of $K_{ATP}$ channels with 100 nM glibenclamide. Upper traces show data from a WT mouse; lower traces show data from a BACHD mouse. (B) Population data (from 5–7-month-old mice). In WT neurons, inhibition of $K_{ATP}$ channels had mixed effects on firing, whereas in BACHD neurons inhibition of $K_{ATP}$ channels increased both the frequency and regularity of firing. (C) Examples of whole-cell recordings from WT and BACHD STN neurons before (left) and after (right) inhibition of $K_{ATP}$ channels with 100 nM glibenclamide. (D) Population data (from 5–7-month-old mice). The interspike voltage trajectory was lower in BACHD neurons compared to WT. $K_{ATP}$ channel inhibition increased the interspike voltage trajectory in BACHD neurons but had no effect in WT. As with cell-attached recordings, inhibition of $K_{ATP}$ channels had mixed effects on firing in WT neurons, whereas in BACHD mice inhibition of $K_{ATP}$ channels increased the frequency and regularity of firing. *p < 0.05. ns, not significant. Data for panel B provided in *Figure 5—source data 1*; data for panel D provided in *Figure 5—source data 2*.

The following source data is available for figure 5:

**Source data 1.** Autonomous firing frequency and CV for WT and BACHD STN neurons under control conditions and following glibenclamide application in *Figure 5B*.

**Source data 2.** Autonomous interspike voltage trajectory, firing frequency and CV for whole-cell recordings from WT and BACHD STN neurons in *Figure 5D*.

increased following application of glibenclamide (BACHD control frequency: 7.7 [4.5–13.4] Hz; BACHD glibenclamide frequency: 15.1 [8.5–20.9] Hz; n = 7; p = 0.0156; BACHD control CV: 0.24 [0.13–0.35]; BACHD glibenclamide CV: 0.13 [0.10–0.15]; n = 7; p = 0.0313; BACHD control interspike voltage trajectory: 125.3 [59.2–298.0] mV/s; BACHD glibenclamide interspike voltage trajectory: 369.9 [156.7–474.0] mV/s; n = 7; $p_h$ = 0.0313; *Figure 5C,D*). In contrast inhibition of $K_{ATP}$ channels did not alter the firing of WT neurons (WT control frequency: 14.7 [12.5–27.4] Hz; WT glibenclamide frequency: 16.8 [8.3–20.3] Hz; n = 7; p = 0.3750; WT control CV: 0.10 [0.07–0.18]; WT glibenclamide CV: 0.08 [0.06–0.16]; n = 7; p = 0.8125; WT control interspike voltage trajectory: 413.8 [317.1–705.0]

mV/s; n = 7; WT glibenclamide interspike voltage trajectory: 361.9 [216.7–522.9] mV/s; n = 7; $p_h$ = 0.3750; *Figure 5C,D*).

Glibenclamide can also activate Epac2 and Rap1 (*Zhang et al., 2009*), which could rescue firing through a pathway that is independent of $K_{ATP}$ channel inhibition. Therefore, gliclazide, a sulfonylurea that has no effect on Epac2/Rap1 signaling (*Zhang et al., 2009*; *Takahashi et al., 2015*), was applied to STN neurons of 5–7-month-old BACHD mice (*Figure 6*). In loose-seal, cell-attached recordings gliclazide (10 µM) increased both firing rate (control: 5.1 [2.0–8.0] Hz; gliclazide: 9.9 [4.6–13.8] Hz; n = 6; p = 0.0313; *Figure 6*) and regularity (control CV: 0.54 [0.31–0.88]; gliclazide CV: 0.29 [0.10–0.47]; n = 6; p = 0.0313; *Figure 6*) in phenotypic BACHD STN neurons. Together, these data argue that $K_{ATP}$ channels are responsible for the impaired autonomous activity of STN neurons in the BACHD model.

As described above, 3–5 hr NMDAR antagonism with D-AP5 partially rescued autonomous activity in BACHD STN neurons. To determine whether this rescue was mediated through effects on $K_{ATP}$ channels, glibenclamide was applied following this treatment. D-AP5 pre-treatment partially occluded the increases in the autonomous firing rate (BACHD glibenclamide Δ frequency: 4.3 [2.2–8.7] Hz, n = 15; D-AP5 pre-treated BACHD glibenclamide Δ frequency: 1.9 [0.7–3.2] Hz, n = 6; p = 0.0365) and regularity (BACHD glibenclamide Δ CV: −0.25 [−0.85− −0.13], n = 14; D-AP5 pre-treated BACHD glibenclamide Δ CV: −0.09 [−0.10− −0.03], n = 6; p = 0.0154) that accompany $K_{ATP}$ channel inhibition. Thus, these observations are consistent with the conclusion that prolonged NMDAR antagonism partially rescued autonomous activity in BACHD STN neurons through a reduction in $K_{ATP}$ channel-mediated firing disruption.

## NMDAR activation produces a persistent $K_{ATP}$ channel-mediated disruption of autonomous activity in WT STN neurons

To further examine whether elevated NMDAR activation can trigger a homeostatic $K_{ATP}$ channel-mediated reduction in autonomous firing in WT STN, brain slices from 2-month-old C57BL/6 mice were incubated in control media or media containing 25 µM NMDA for 1 hr prior to recording (*Figure 7*). NMDA pre-treatment reduced the proportion of autonomously firing neurons (untreated: 66/75 (88%); NMDA: 65/87 (75%); p = 0.0444) and the frequency (untreated: 14.9 [7.8–24.8] Hz; n = 75; NMDA: 5.2 [0.0–14.0] Hz; n = 87; $p_h$ < 0.0001) and regularity (untreated CV: 0.13 [0.08–0.25]; n =

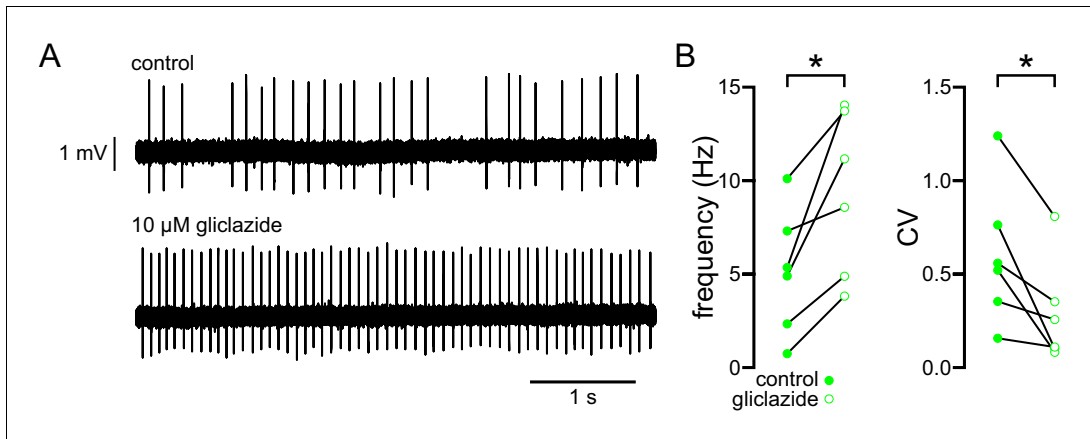

**Figure 6.** The abnormal autonomous activity of STN neurons in BACHD mice is rescued by inhibition of $K_{ATP}$ channels with gliclazide. (**A**) Examples of loose-seal cell-attached recordings of a STN neuron from a 6-month-old BACHD mouse before (upper) and after (lower) inhibition of $K_{ATP}$ channels with 10 µM gliclazide. (**B**) Population data (5–7-month-old). In BACHD STN neurons inhibition of $K_{ATP}$ channels with gliclazide increased the frequency and regularity of firing. *p < 0.05. Data for panel **B** provided in *Figure 6—source data 1*.

The following source data is available for figure 6:

**Source data 1.** Autonomous firing frequency and CV for WT and BACHD STN neurons under control conditions and following gliclazide application in *Figure 6B*.

66; NMDA CV: 0.24 [0.10–0.72]; n = 65; $p_h$ = 0.0150; *Figure 7A–C*) of autonomous activity relative to control slices. In a subset of neurons glibenclamide was applied to inhibit $K_{ATP}$ channels. In neurons from untreated slices glibenclamide had no effect on firing rate (control: 16.6 [10.9–31.3] Hz; glibenclamide: 25.0 [16.3–32.8] Hz; n = 6; $p_h$ = 0.2188; *Figure 7A–D*) or regularity (control CV: 0.08 [0.07–0.37]; glibenclamide CV: 0.08 [0.06–0.09]; n = 6; $p_h$ = 0.3125; *Figure 7A–D*). However, in neurons from NMDA pre-treated slices glibenclamide application elevated firing rate (control: 3.3 [2.3–5.1] Hz; glibenclamide: 11.4 [10.8–24.4] Hz; n = 10; $p_h$ = 0.0078; *Figure 7A–D*) and regularity (control CV: 0.83 [0.25–1.03]; glibenclamide CV: 0.12 [0.07–0.16]; n = 8, $p_h$ = 0.0208; *Figure 7A–D*) to levels similar to that seen in untreated slices. Together, these data demonstrate that increased activation of STN NMDARs can lead to a persistent $K_{ATP}$ channel-mediated homeostatic reduction in autonomous activity in STN neurons.

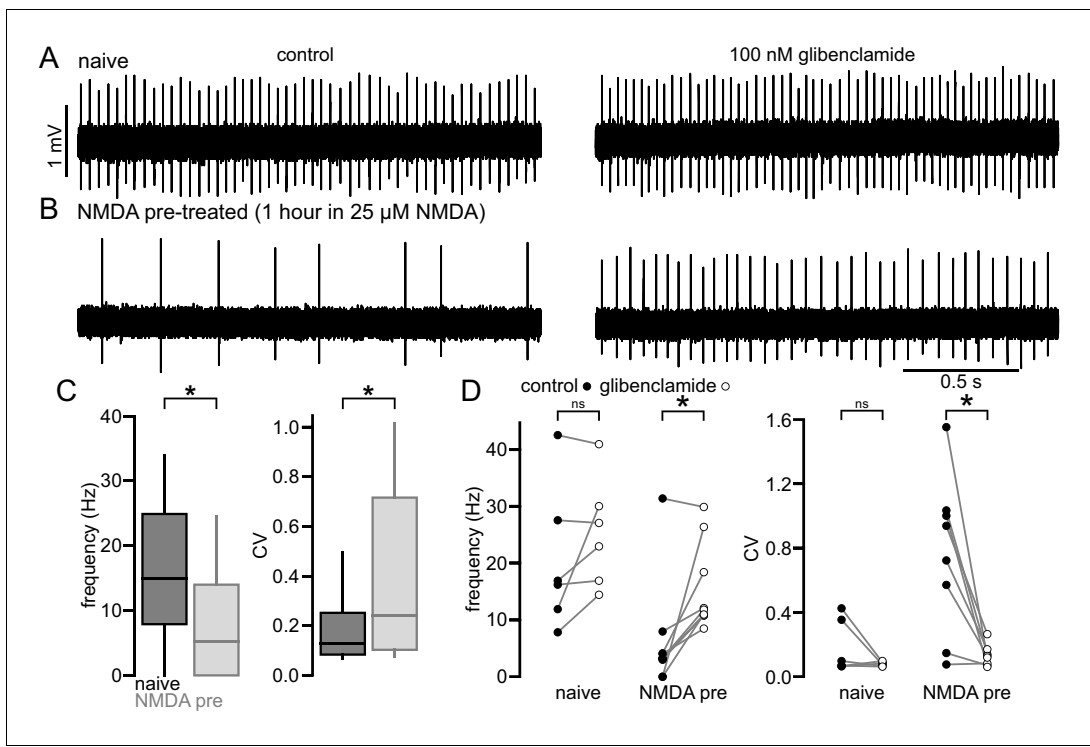

**Figure 7.** NMDA exposure persistently disrupts autonomous activity. (**A**) Example of a loose-seal cell-attached recording from an STN neuron from an untreated slice from a C57/BL6 mouse before (left) and after (right) inhibition of $K_{ATP}$ channels with 100 nM glibenclamide. (**B**) Example of a loose-seal cell-attached recording from an STN neuron from a slice treated with 25 µM NMDA for 1 hr prior to recording, before (left) and after (right) inhibition of $K_{ATP}$ channels with 100 nM glibenclamide. (**C**) Population data. Following NMDA pre-treatment STN neurons had a lower frequency and regularity of firing. (**D**) Inhibition of $K_{ATP}$ channels with 100 nM glibenclamide restored firing in STN cells from NMDA pre-treated slices but had no effect on firing from naïve slices. *p < 0.05. ns, not significant. Data for panel **C** provided in *Figure 7—source data 1*; data for panel **D** provided in *Figure 7—source data 2*.

The following source data is available for figure 7:

**Source data 1.** Autonomous firing frequency and CV for control and NMDA pre-treated C57BL/6 STN neurons in *Figure 7C*.

**Source data 2.** Autonomous firing frequency and CV for control and NMDA pre-treated C57BL/6 STN neurons in control conditions and following glibenclamide application in *Figure 7D*.

## Break down of $H_2O_2$ rescues the firing of BACHD STN neurons to WT levels

Mitochondrial oxidative phosphorylation generates superoxide, which may be dismuted by superoxide dismutase to produce $H_2O_2$ (*Adam-Vizi, 2005*). Superoxide and hydrogen peroxide can also be produced by NADPH oxidase (*Brennan et al., 2009*). Because $K_{ATP}$ channels are activated by $H_2O_2$ (*Ichinari et al., 1996*; *Avshalumov et al., 2005*), we tested whether $H_2O_2$ contributes to $K_{ATP}$ channel-mediated disruption of ex vivo autonomous activity in the BACHD model. The effect of a membrane permeable form of the enzyme catalase (polyethylene glycol-catalase), which breaks down $H_2O_2$, on the autonomous firing of STN neurons from 4–6-month-old BACHD mice was examined (*Figure 8*). Application of 250 U/ml catalase increased the rate (BACHD control: 3.4 [0.7–5.5] Hz; BACHD catalase: 10.8 [7.6–13.8] Hz; n = 11; $p_h$ = 0.0080; *Figure 8A–C*) and regularity (BACHD control CV: 1.0 [0.3–2.1]; BACHD catalase CV: 0.17 [0.12–0.21]; n = 11; $p_h$ = 0.0060; *Figure 8A–C*) of autonomous firing. Subsequent application of glibenclamide (100 nM) had no additional effect on firing rate (11.8 [8.2–14.4] Hz; n = 11; $p_h$ = 0.9658) or regularity (CV: 0.15 [0.12–0.23]; n = 11; $p_h$ = 0.4922; *Figure 8A–C*). Thus, these results suggest that $H_2O_2$ underlies $K_{ATP}$ channel activation in BACHD STN neurons.

To test if the actions of $H_2O_2$ on autonomous firing are confined to its effects on $K_{ATP}$ channels, these experiments were repeated in the presence of glibenclamide. As seen previously, application of glibenclamide increased firing rate (BACHD control: 5.2 [1.0–6.7] Hz; BACHD glibenclamide: 8.5 [7.2–11.6] Hz; n = 8, $p_h$ = 0.0156; *Figure 8D*) and regularity (BACHD control CV: 0.83 [0.27–1.30]; BACHD glibenclamide CV: 0.23 [0.15–0.58]; n = 8; $p_h$ = 0.0156; *Figure 8D*). Subsequent application of catalase had no additional effect on firing rate (8.7 [7.2–14.1] Hz; n = 8; $p_h$ = 0.6406; *Figure 8D*) but did produce a small but statistically significant increase in regularity (CV: 0.14 [0.11–0.21]; n = 8; $p_h$ = 0.0156; *Figure 8D*). In WT mice, catalase application did not change firing rate (WT control: 11.0 [10.5–14.2] Hz; WT catalase: 14.3 [11.3–18.3] Hz; n = 7; p = 0.0781; *Figure 9*) but lead to a small but statistically significant increase in regularity (WT control CV: 0.12 [0.10–0.23]; WT catalase CV: 0.11 [0.07–0.13]; n = 7; p = 0.0469; *Figure 9*). The effects of catalase on the frequency and regularity of firing in BACHD mice were greater than those observed in WT mice (frequency: p = 0.0154; CV: p = 0.0007; *Figure 9*). Together, these data suggest that suppression of autonomous activity of STN neurons in BACHD mice is largely mediated by the modulatory effect of $H_2O_2$ on $K_{ATP}$ channels.

To test if the elevation of oxidant stress can result in $K_{ATP}$ channel activation in WT STN neurons, glutathione peroxidase was inhibited with mercaptosuccinic acid (MCS) (*Avshalumov et al., 2005*). Following the application of 1 mM MCS both the rate (control: 12.0 [7.8–13.5] Hz; MCS: 9.0 [4.8–10.5] Hz; n = 11; $p_h$ = 0.0068; *Figure 10*) and regularity (control CV: 0.21 [0.15–0.22]; MCS CV: 0.30 [0.22–0.34]; n = 11; $p_h$ = 0.0137) of firing decreased. Subsequent application of glibenclamide rescued both firing rate (14.6 [10.3–19.2] Hz; n = 11; $p_h$ = 0.0020) and regularity (CV: 0.12 [0.12–0.17]; n = 11; $p_h$ = 0.0098; *Figure 10*). These data are also consistent with an action of $H_2O_2$ on STN $K_{ATP}$ channels.

## The STN degenerates progressively in BACHD mice

HD patients exhibit 20–30% STN neuron loss (*Lange et al., 1976*; *Guo et al., 2012*). Because mitochondrial oxidant stress and reactive oxygen species can trigger apoptotic pathways leading to cell death (*Green and Reed, 1998*; *Bossy-Wetzel et al., 2008*), the number of STN neurons in 12-month-old BACHD and WT mice were compared (*Figure 11*). The brains of BACHD mice and WT littermates were first fixed by transcardial perfusion of formaldehyde, sectioned into 70 μm coronal slices and immunohistochemically labeled for neuronal nuclear protein (NeuN). The total number of NeuN-immunoreactive STN neurons and the volume of the STN were then estimated using unbiased stereological techniques. Both the total number of STN neurons (WT: 10,793 [9,070–11,545]; n = 7; BACHD: 7,307 [7,047–9,285]; n = 7; p = 0.0262) and the volume of the STN (WT: 0.087 [0.084–0.095] $mm^3$; n = 7; BACHD: 0.078 [0.059–0.081] $mm^3$; n = 7; p = 0.0111; *Figure 11A,B*) were reduced in 12-month-old BACHD versus WT mice. The density of STN neurons was not different in BACHD and WT mice (WT: 121,248 [107,180–126,139] neurons/$mm^3$; n = 7; BACHD: 115,273 [90,377–135,765] neurons/$mm^3$; n = 7; p = 0.8048; *Figure 11A,B*). To determine whether the difference in cell number represents an early developmental abnormality or a progressive loss of adult

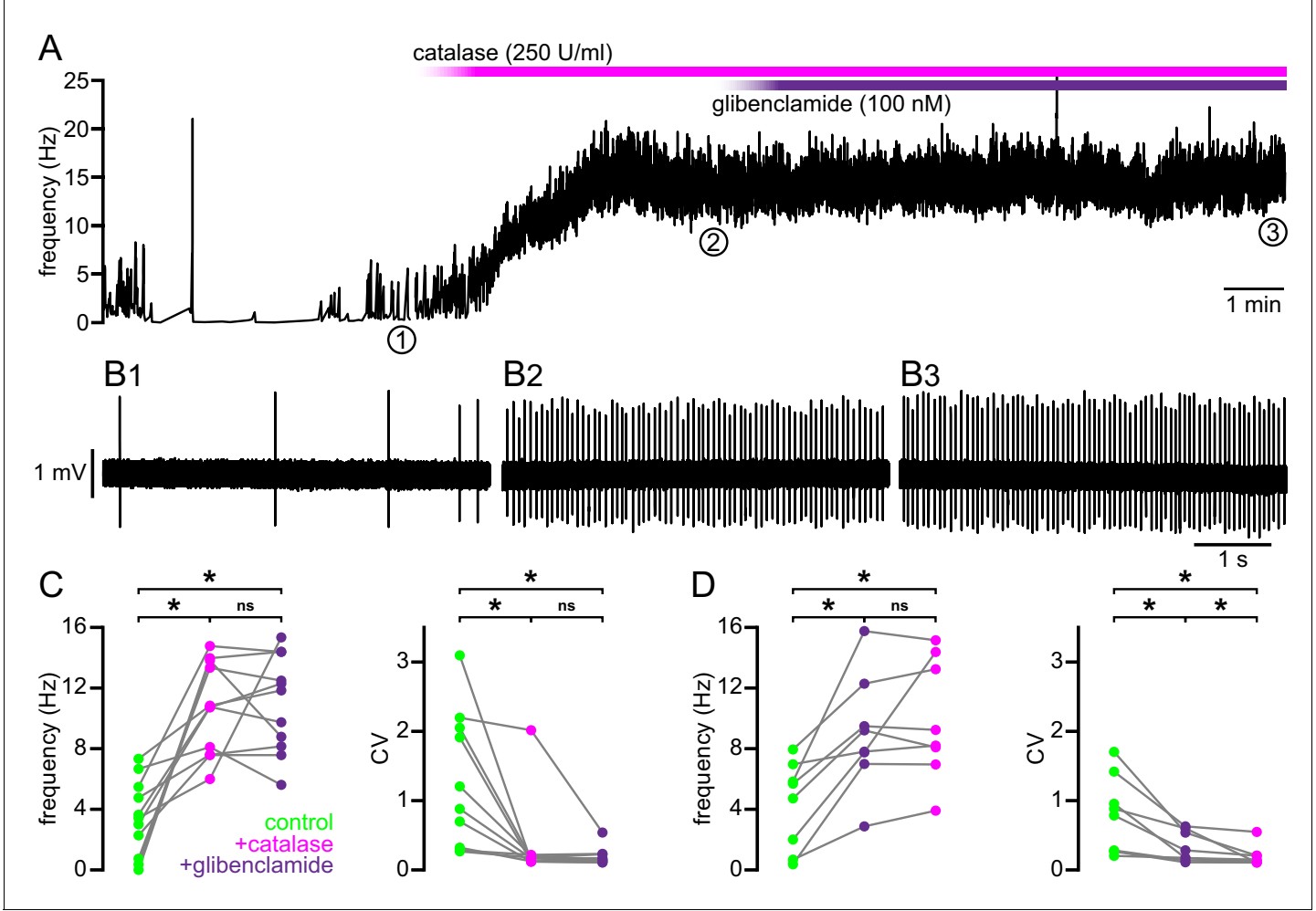

**Figure 8.** Break down of $H_2O_2$ by catalase rescues autonomous firing in BACHD STN neurons. (**A**) Example showing the instantaneous firing rate of a BACHD STN neuron in control conditions, during the application of catalase (250 U/ml), and during co-application of catalase and glibenclamide (100 nM). (**B1**) Example of BACHD STN neuron firing in control conditions (marked 1 in **A**). (**B2**) Example of elevated firing during break down of $H_2O_2$ by catalase (marked 2 in **A**). (**B3**) Example showing no further elevation of firing rate during additional inhibition of $K_{ATP}$ channels with glibenclamide (marked 3 in **A**). (**C**) Population data from 4–6-month old BACHD mice showing an increase in the frequency and regularity of firing following break down of $H_2O_2$, with no further changes upon $K_{ATP}$ channel inhibition. (**D**) Population data showing an increase in the frequency and regularity of firing following $K_{ATP}$ channel inhibition with no further change in firing rate and a slight increase in firing regularity upon $H_2O_2$ break down. *$p < 0.05$. ns, not significant. Data for panels **C**–**D** provided in *Figure 8—source data 1*.

The following source data is available for figure 8:

**Source data 1.** Autonomous firing frequency and CV for WT and BACHD STN neurons under control conditions and following catalase and/or glibenclamide application in *Figure 8C–D*.

neurons, the numbers of neurons in 2-month-old BACHD and WT mice were also compared. At 2-months-old, the total number of STN neurons (WT: 10,373 [9,341–14,414]; n = 7; BACHD: 10,638 [10,513–13,877]; n = 7; p = 0.7104; *Figure 11C*), the volume of the STN (WT: 0.098 [0.090–0.125] mm$^3$; n = 7; BACHD: 0.085 [0.080–0.111] mm$^3$; n = 7; p = 0.1649; *Figure 11C*) and STN neuronal density (106,880 [98,100–115,985] neurons/mm$^3$; n = 7; BACHD: 124,844 [115,479–145,711] neurons/mm$^3$; n = 7; p = 0.1282; *Figure 11C*) were not different in WT and BACHD mice. Together, these data demonstrate that between the ages of 2 months and 12 months BACHD mice lose approximately one third of their STN neurons compared to WT littermates.

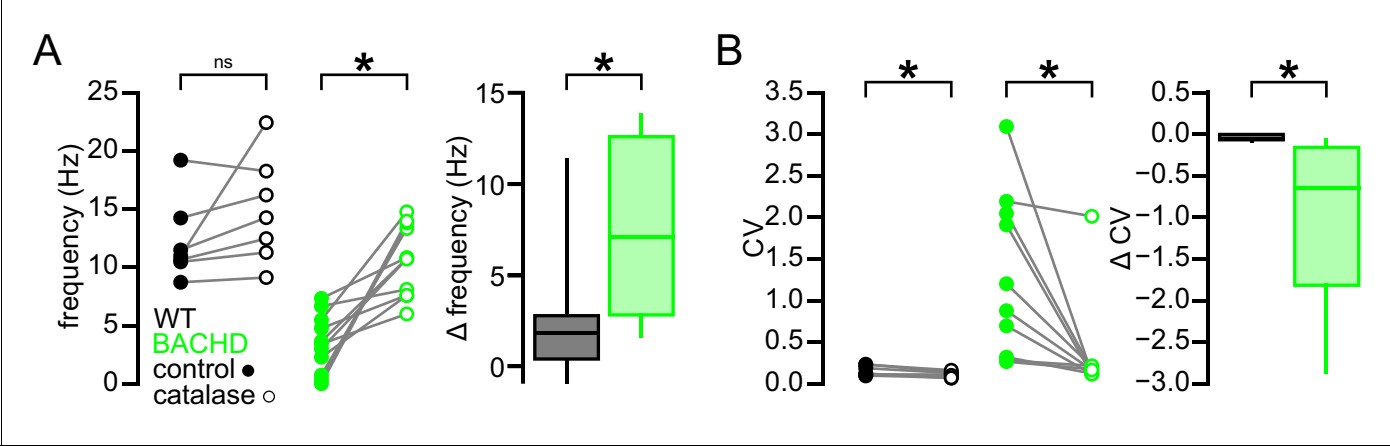

**Figure 9.** Break down of $H_2O_2$ by catalase has a relatively minimal effect on autonomous firing in WT STN neurons compared to BACHD neurons. (**A**) Line plots showing of the effect of catalase (250 U/ml) on the frequency of autonomous action potential generation in STN neurons from WT (black) and BACHD mice (green; BACHD data same as in *Figure 8C*). Break down of $H_2O_2$ elevated autonomous firing in BACHD STN neurons only. The boxplot confirms that the elevation of firing due to catalase application was greater in BACHD mice. (**B**) Line plots illustrating a small but statistically significant effect of catalase on the regularity of autonomous action potential generation in STN neurons from WT mice (black) compared to a larger increase in regularity following catalase application in BACHD neurons (green; BACHD data same as in *Figure 8C*). The boxplot confirms that the increase in regularity due to catalase was greater in BACHD mice. *p < 0.05. ns, not significant. Data provided in *Figure 9—source data 1*.

The following source data is available for figure 9:

**Source data 1.** Autonomous firing frequency and CV for WT and BACHD STN neurons under control conditions and following catalase application in *Figure 9*.

## The STN of Q175 KI mice exhibits similar abnormalities to those observed in the BACHD model

STN neurons from BACHD mice exhibit perturbed autonomous firing that is caused by NMDAR activation/signaling leading to mitochondrial oxidant stress, $H_2O_2$ generation and $K_{ATP}$ channel activation. Furthermore, STN neurons are progressively lost in BACHD mice. To determine whether these features are specific to the BACHD model or a more general feature of HD models, a subset of experiments were repeated in heterozygous Q175 KI mice (*Figure 12*). STN neurons from 6-month-old Q175 mice exhibited a severely reduced rate of autonomous activity (WT: 7.8 [1.9–14.7] Hz; n = 90; Q175: 0.0 [0.0–6.3] Hz; n = 90; p < 0.0001; *Figure 12A,B*), though the regularity of active neurons was unchanged (WT CV: 0.2 [0.1–0.6]; n = 77; Q175 CV: 0.4 [0.1–1.0]; n = 42; p = 0.1506; *Figure 12A,B*). Additionally, there was a large decrease in the proportion of active neurons in the Q175 STN (WT: 77/90 (86%); Q175: 42/90 (47%); p < 0.0001).

Inhibition of $K_{ATP}$ channels with glibenclamide rescued both STN firing rate and regularity in Q175 and increased regularity only in WT (WT control frequency: 9.7 [5.4–13.5] Hz; WT glibenclamide frequency: 10.3 [7.4–15.4] Hz; n = 8; p = 0.1094; Q175 control frequency: 4.8 [3.5–6.2] Hz; Q175 glibenclamide frequency: 11.0 [9.3–13.6] Hz; n = 6; p = 0.0313; WT control CV: 0.19 [0.13–0.47]; WT glibenclamide CV: 0.11 [0.10–0.21]; n = 8; p = 0.0078; Q175 control CV: 0.45 [0.35–0.71]; Q175 glibenclamide CV: 0.15 [0.10–0.17]; n = 6; p = 0.03125; *Figure 12C,D*). Similar to BACHD, Q175 STN neurons recovered to WT-like firing rate following >3 hr pretreatment with D-AP5 (Q175 control: 4.6 [0.0–11.4] Hz; n = 45; Q175 D-AP5 treated: 11.6 [0.0–18.7] Hz; n = 45; p = 0.0144; *Figure 12E,F*), although the regularity (Q175 control CV: 0.16 [0.10–0.66]; n = 15; Q175 D-AP5 treated CV: 0.14 [0.09–0.32]; n = 12; p = 0.2884; *Figure 12E,F*) and proportion of active neurons (Q175 control: 30/45 (67%); Q175 D-AP5 treated: 33/45 (73%); p = 0.6460; *Figure 12E,F*) were unaltered. The 12-month-old Q175 STN (n = 7) exhibited a median 26% reduction in the total number of STN neurons with no effect on other parameters (WT: 8,661 [7,120–9,376] neurons; Q175: 6,420 [5,792–7,024] neurons; p = 0.0111; WT volume: 0.081 [0.074–0.087] $mm^3$; Q175 volume: 0.079 [0.070–0.091] $mm^3$; p = 0.6200; WT density: 109,477 [82,180–115,301] neurons/$mm^3$; Q175 density: 88,968

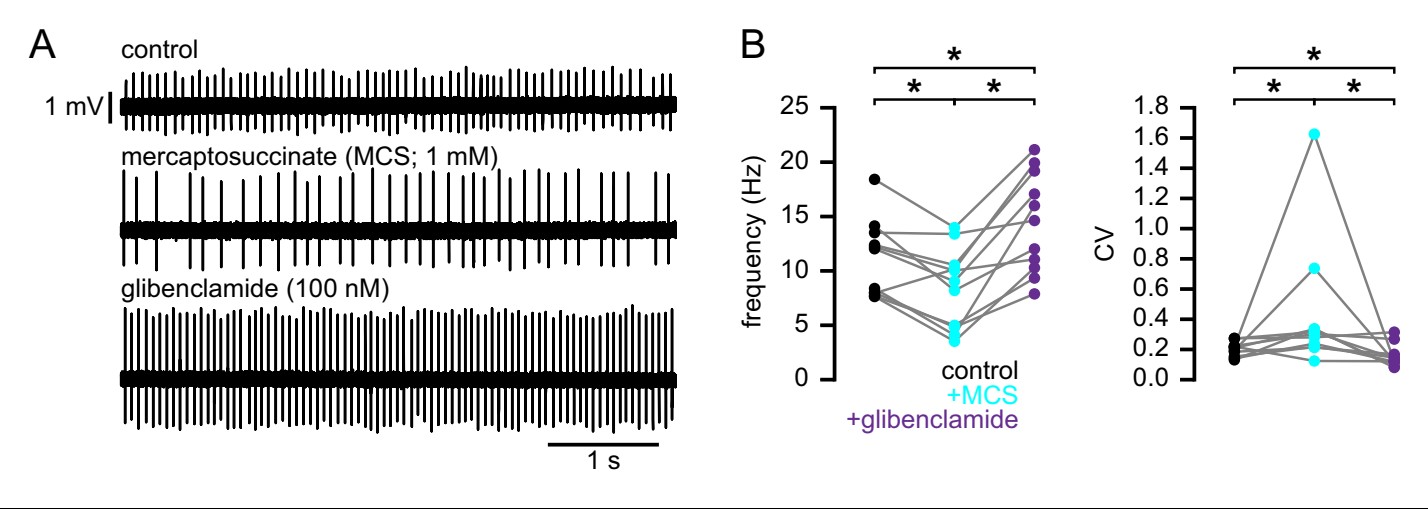

**Figure 10.** Increasing $H_2O_2$ levels by inhibition of glutathione peroxidase with mercaptosuccinic acid in WT mice leads to disruption of autonomous action potential generation through activation of $K_{ATP}$ channels. (**A**) Example of autonomous activity of a STN neuron from a C57BL/6 mouse in control conditions (upper), during application of 1 mM mercaptosuccinic acid (MCS; middle), and during subsequent application of 100 nM glibenclamide (lower). These recordings were made in the presence of 20 μM flufenamic acid to block transient receptor potential (TRP) channels (*Lee et al., 2011*). (**B**) Population data showing a decrease in the frequency and regularity of firing following MCS application, which was reversed by subsequent $K_{ATP}$ channel inhibition. *p < 0.05. Data for panel **B** provided in *Figure 10—source data 1*.

The following source data is available for figure 10:

**Source data 1.** Autonomous firing frequency and CV for WT and BACHD STN neurons under control conditions and following MCS and glibenclamide application in *Figure 10B*.

[63,624–103,020] neurons/mm$^3$; p = 0.2086; *Figure 12G,H*). Taken together, these data show that the STN exhibits similar dysfunction and neuronal loss in both the transgenic BACHD and Q175 KI mouse models of HD.

## Discussion

Dysfunction of the striatum and cortex has been extensively characterized in HD models, but relatively few studies have examined the extra-striatal basal ganglia. Here, we report early NMDAR, mitochondrial and firing abnormalities together with progressive loss of STN neurons in two HD mouse models. Furthermore, dysfunction was present in HD mice prior to the onset of major symptoms, implying that it occurs early in the disease process (*Gray et al., 2008*; *Menalled et al., 2012*). Cell death in the STN also preceded that in the striatum, as STN neuronal loss was observed at 12 months of age in both BACHD and Q175 mice, a time point at which striatal neuronal loss is absent but psychomotor dysfunction is manifest (*Gray et al., 2008*; *Heikkinen et al., 2012*; *Smith et al., 2014*; *Mantovani et al., 2016*). Together these findings argue that dysfunction within the STN contributes to the pathogenesis of HD.

Astrocytes appear to play a pivotal role in HD. Expression of mutant huntingtin in astrocytes alone is sufficient to recapitulate neuronal and neurological abnormalities observed in HD and its models (*Bradford et al., 2009*; *Faideau et al., 2010*). Furthermore, astrocyte-specific rescue approaches ameliorate some of the abnormalities observed in HD models (*Tong et al., 2014*; *Oliveira et al., 2016*). In the STN, inhibition of GLT-1 (and GLAST) slowed individual NMDAR EPSCs in WT but not BACHD mice and eliminated the differences in their decay kinetics, arguing that impaired uptake of glutamate by astrocytes contributed to the relative prolongation of NMDAR-mediated EPSCs in BACHD STN neurons. Interestingly, and in contrast to the striatum (*Milnerwood et al., 2010*), when spillover of glutamate onto extrasynaptic receptors was increased by train stimulation and inhibition of astrocytic glutamate uptake, the resulting compound NMDAR EPSC and its prolongation by uptake inhibition were similar in BACHD and WT mice, arguing against

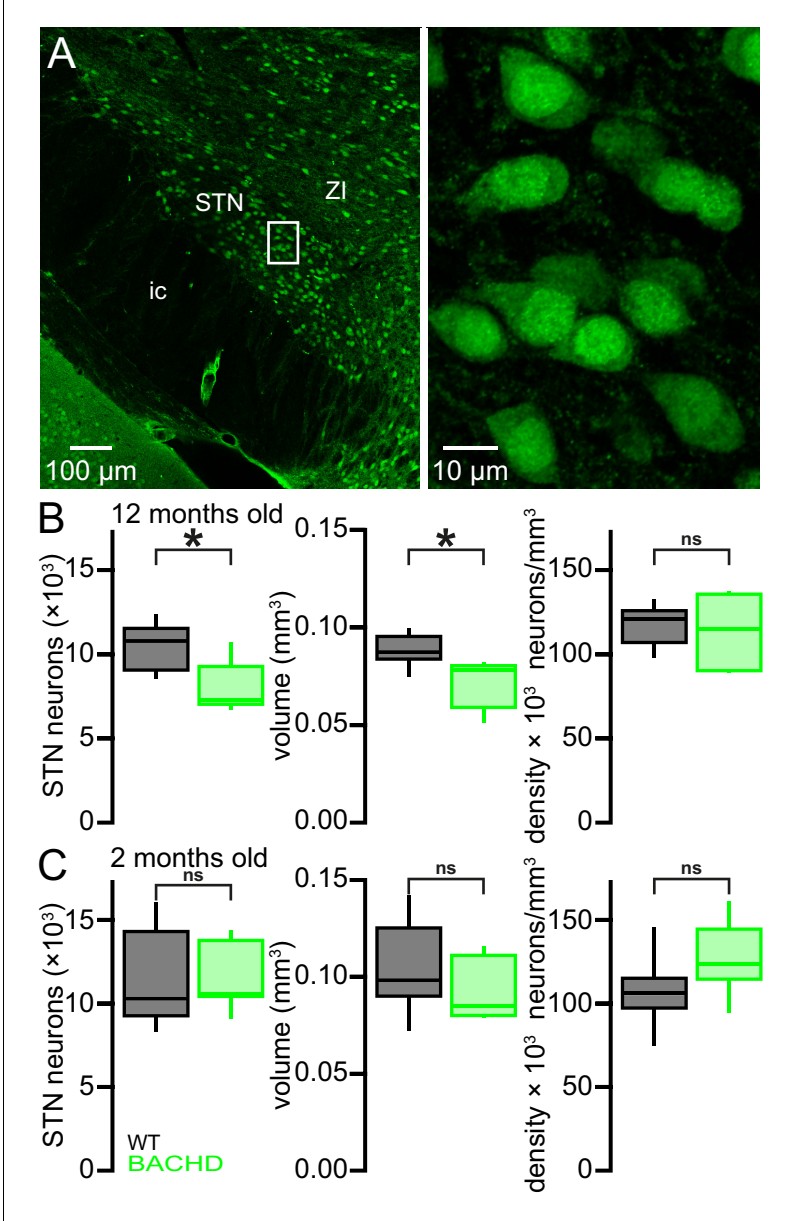

**Figure 11.** Degeneration of STN neurons in BACHD mice. (**A**) Expression of NeuN in STN neurons in a BACHD mouse (ZI, zone incerta; ic, internal capsule). (**B**) Population data showing a 32.3% reduction in the median STN neuron number and a 10.3% reduction in STN volume at 12 months old. *p < 0.05. ns, not significant. (**C**) Population data showing no difference in STN neuron number, STN volume or density between WT and BACHD mice at 2 months old. Data for panels **B**–**C** provided in *Figure 11—source data 1*.

The following source data is available for figure 11:

**Source data 1.** BACHD STN neuron counts, density and STN volume in *Figure 11B–C*.

an increase in extrasynaptic STN NMDAR expression/function in BACHD mice. Slowing of astrocytic glutamate uptake has recently been shown to occur during increased presynaptic activity (*Armbruster et al., 2016*). Thus, train stimulation may have slowed glutamate uptake sufficiently to occlude/eliminate the differences in uptake that were observed in BACHD and WT STN neurons during single stimulation. Whether the modest differences in glutamate uptake that

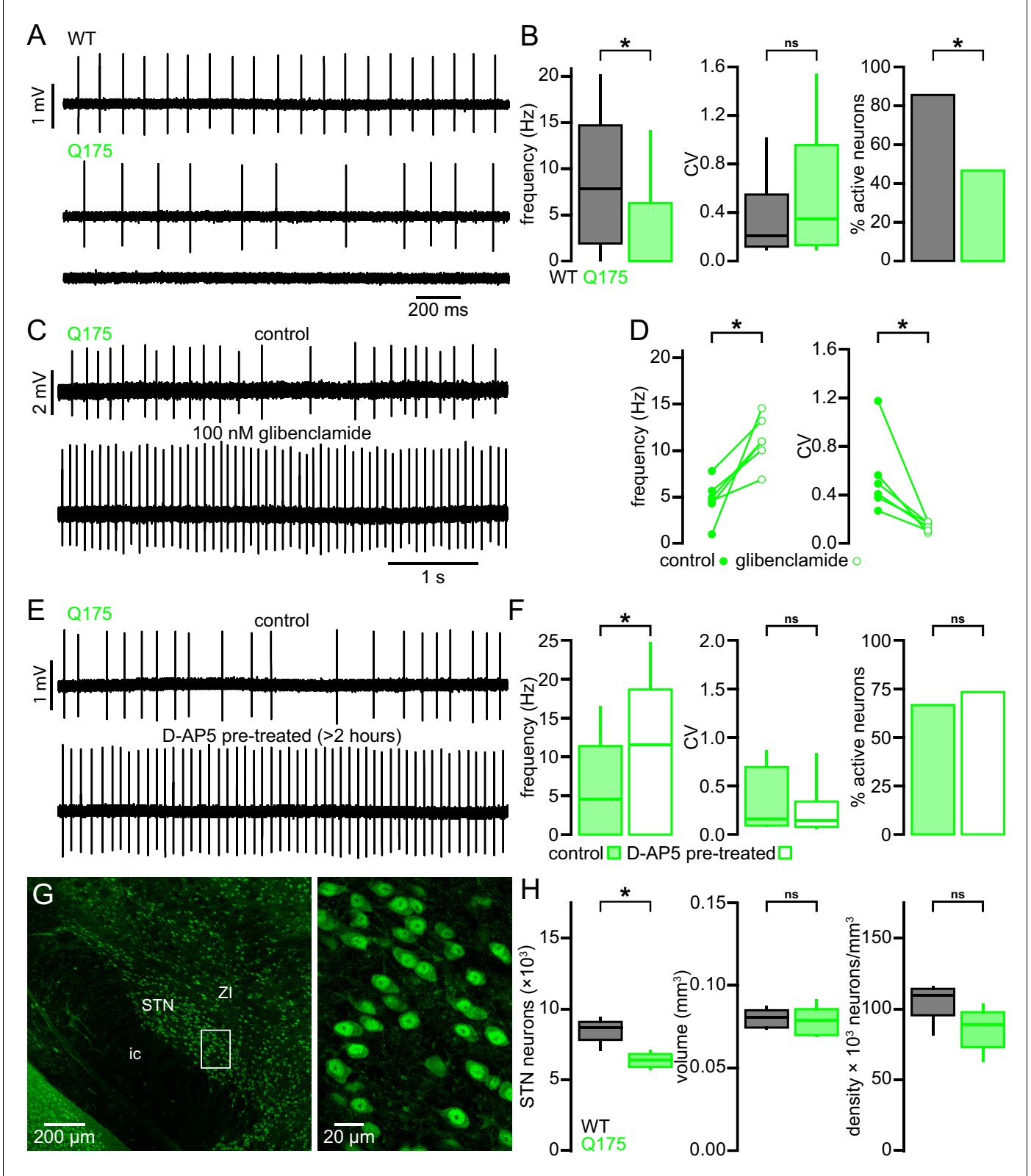

**Figure 12.** STN neurons exhibit similar dysfunction and degeneration in Q175 mice. (**A**) Examples of impaired (middle, lower) autonomous firing of neurons from Q175 mice relative to WT (upper). (**B**) Population data showing STN neuron firing rate, CV, and proportion of active neurons from 6-month-old WT and Q175 mice. (**C**) Example showing the autonomous firing of a STN neuron from a Q175 mouse before (upper) and during (lower) inhibition of $K_{ATP}$ channels with glibenclamide (100 nM). (**D**) Paired data confirming that inhibition of $K_{ATP}$ channels in Q175 STN neurons consistently

*Figure 12 continued on next page*

*Figure 12 continued*

increased the rate (left) and regularity (right) of autonomous firing. (**E**) Examples of autonomous action potential generation from Q175 STN neurons from an untreated slice (upper) and a slice treated with 50 µM D-AP5 for 3–5 hr prior to recording (lower). (**F**) Population data showing STN neuron firing rate, CV, and the proportion of active cells in untreated and D-AP5 treated slices. The frequency of autonomous firing increased in D-AP5 treated slices. (**G**) Expression of NeuN in STN neurons in a Q175 mouse. (**H**) Population data showing STN neuron number (left), STN volume (center), and neuron density (right) in 12-month-old Q175 mice. The number of STN neurons was lower in Q175 mice compared to WT. *p < 0.05. ns, not significant. Data for panel B provided in *Figure 12—source data 1*; data for panel D provided in *Figure 12—source data 2*; data for panel F provided in *Figure 12—source data 3*; data for panel H provided in *Figure 12—source data 4*.

The following source data is available for figure 12:

**Source data 1.** Autonomous firing frequency and CV for Q175 and WT STN neurons in *Figure 12B*.

**Source data 2.** Autonomous firing frequency and CV for Q175 in control conditions and following glibenclamide application *Figure 12D*.

**Source data 3.** Autonomous firing frequency and CV for control and D-AP5 pre-treated Q175 STN neurons in *Figure 12F*.

**Source data 4.** Q175 STN neuron counts, density and STN volume in *Figure 12H*.

were observed here are sufficient to promote NMDAR-mediated dysfunction in HD STN neurons remains to be determined.

NMDARs play a key role in the abnormal activity of STN neurons in HD models. Antagonism of STN NMDARs in BACHD and Q175 brain slices rescued autonomous STN firing. Conversely, acute activation of STN NMDARs persistently disrupted STN firing in WT brain slices. If the relatively low level of glutamatergic transmission present ex vivo is sufficient to impair firing then this impairment is likely to be more severe in vivo where STN neurons are powerfully patterned by glutamatergic transmission arising from the cortex, thalamus, pedunculopontine nucleus and superior colliculus (reviewed by *Bevan, 2017*). Non-synaptic sources of extracellular glutamate, such as diffusion/release from astrocytes (*Cavelier and Attwell, 2005*; *Lee et al., 2013*) may also contribute to excessive NMDAR activation in HD mice.

Extended antagonism of NMDARs in BACHD slices also reduced mitochondrial oxidant stress in STN neurons. NMDAR activation can elevate ROS through a variety of $Ca^{2+}$- and nitric oxide-associated signaling pathways and their actions on mitochondria, NADPH oxidase and antioxidant expression (*Dugan et al., 1995*; *Moncada and Bolaños, 2006*; *Brennan et al., 2009*; *Nakamura and Lipton, 2011*; *Valencia et al., 2013*). Although we saw no evidence of basal mitochondrial dysfunction that was not attributable to increased NMDAR function, there is considerable evidence that mutant huntingtin causes transcriptional dysregulation, which leads to defective mitochondrial quality control, an increase in the proportion of defective, ROS generating mitochondria and an increase in opening of the permeability transition pore (*Milakovic and Johnson, 2005*; *Panov et al., 2002*; *Fernandes et al., 2007*; *Song et al., 2011*; *Chaturvedi et al., 2013*; *Johri et al., 2013*; *Martin et al., 2015*). Thus, basal mitochondrial dysfunction could render HD STN neurons especially sensitive to NMDAR-mediated transmission and signaling.

Catalase rapidly restored autonomous firing in the BACHD model, an effect occluded by inhibition of $K_{ATP}$ channels, arguing that $H_2O_2$, through its action on $K_{ATP}$ channels is the major cause of firing disruption. $H_2O_2$ can act on $K_{ATP}$ channels by decreasing their sensitivity to ATP (*Ichinari et al., 1996*), reducing the ratio of ATP to ADP (*Krippeit-Drews et al., 1999*), and/or modulating channel gating through a sGC-cGMP-PKG-ROS($H_2O_2$)-ERK1/2-calmodulin-CaMKII signaling pathway (*Zhang et al., 2014*). $H_2O_2$ is likely to directly modulate STN $K_{ATP}$ channels in HD mice because disrupted firing was also observed when STN neurons were recorded in the whole-cell configuration with patch pipettes containing exogenous ATP. Furthermore, $H_2O_2$ break down rapidly rescued activity, consistent with a direct action on $K_{ATP}$ channels. $H_2O_2$-dependent modulation of $K_{ATP}$ channels has been extensively characterized in midbrain dopamine neurons where it powerfully suppresses cellular excitability and synaptic transmission (*Avshalumov et al., 2005*; *Bao et al., 2009*). The activation of $K_{ATP}$ channels in STN neurons may represent a form of homeostasis that suppresses firing when mitochondrial oxidant stress is high, limiting the possibility of oxidant damage and bioenergetic failure (*Ray et al., 2012*; *Sena and Chandel, 2012*).

In HD, chronic oxidant stress can lead to damage, such as lipid and protein peroxidation and nuclear/mitochondrial DNA damage, which profoundly impair cellular function and promote cell death (*Perluigi et al., 2005*; *Browne and Beal, 2006*; *Acevedo-Torres et al., 2009*). Consistent with the negative effects of such processes on neuronal viability, we observed progressive loss of STN neurons in both the BACHD and Q175 models. Furthermore, the level of neuronal loss at 12 months in the BACHD and Q175 models was similar to that observed in HD patients (*Lange et al., 1976*; *Guo et al., 2012*). The absence of neuronal loss in the cortex and striatum in the same models at an equivalent time point suggests that STN dysfunction and degeneration may be particularly influential in the early disease process. Although the STN is known to degenerate in HD, it is not clear why neuronal loss is ultimately less than that observed in the striatum at the end stage of the disease, despite the fact that dysfunction and degeneration occur earlier (at least in HD models). Future research will be required to determine whether subtypes of STN neurons exhibit selective vulnerability and/or whether the processes promoting their degeneration, e.g. cortical activation of STN NMDARs, ultimately wane.

As a key component of the hyperdirect and indirect pathways, the STN is critical for constraining cortico-striatal activity underlying action selection (*Albin et al., 1989*; *Oldenburg and Sabatini, 2015*). In the 'classical' model of basal ganglia function, degeneration of indirect pathway striatal projection neurons is proposed to underlie the symptoms of early stage HD (*Albin et al., 1989*). Here we show for the first time that STN dysfunction and neuronal loss precede cortico-striatal abnormalities in HD models. Thus, dysfunction and degeneration of cortical and striatal neurons occurs in concert with profound changes in other elements of the basal ganglia. Therapeutic strategies that target the STN may therefore be useful not only for treating the psychomotor symptoms of early- to mid-stage HD but also for influencing dysfunction and degeneration throughout the cortico-basal ganglia-thalamo-cortical circuit.

## Materials and methods

### Animals

All animal procedures were performed in accordance with the policies of the Society for Neuroscience and the National Institutes of Health, and approved by the Institutional Animal Care and Use Committee of Northwestern University. Adult male hemizygous BACHD mice (RRID:IMSR_JAX: 008197) and heterozygous Q175 mice (RRID:IMSR_JAX:027410), their WT litter mates, and C57BL/6 mice (Charles River Laboratories International, Inc., Wilmington, MA, USA) were used in this study.

### Stereotaxic injection of viral vectors

Mice were anesthetized with 1–2% isoflurane (Smiths Medical ASD, Inc., Dublin, OH, USA). AAV vectors (serotype 9; ~$10^{12-13}$ GC/ml) engineered to express hChR2(H134R)-eYFP under the hSyn promoter (University of Pennsylvania Vector Core, Philadelphia, PA, USA) or MTS-roGFP under the CMV promoter (*Sanchez-Padilla et al., 2014*) were injected under stereotaxic guidance (Neurostar, Tubingen, Germany). In order to express hChR2(H134R)-eYFP, AAV was injected bilaterally into the primary motor cortex (three injections per hemisphere; coordinates relative to bregma: AP, +0.6 mm, + 1.2 mm, and +1.8 mm; ML, + 1.5 mm, and −1.5 mm; DV, −1.0 mm; 0.3 µl per injection). In order to express MTS-roGFP, AAV was injected bilaterally into the STN (coordinates: AP, −2.06 mm; ML, +1.4 mm, and −1.4 mm; DV, −4.5 mm; 0.4 µl per injection). Brain slices were prepared from AAV-injected mice 10–14 days after injection.

### Slice preparation

Mice were lightly anesthetized with isoflurane, deeply anesthetized with ketamine/xylazine (87/13 mg/kg i.p.) and then perfused transcardially with ~10 ml of ice-cold sucrose-based artificial cerebrospinal fluid (sACSF) that contained 230 mM sucrose, 2.5 mM KCl, 1.25 mM $NaH_2PO_4$, 0.5 mM $CaCl_2$, 10 mM $MgSO_4$, 10 mM glucose, and 26 mM $NaHCO_3$ equilibrated with 95% $O_2$ and 5% $CO_2$. The brain was removed, immersed in sASCF and 250 µm sagittal slices were cut with a vibratome (VT1200S; Leica Microsystems Inc., IL, USA). Slices were then transferred to a holding chamber, immersed in ACSF that contained 125 mM NaCl, 2.5 mM KCl, 1.25 mM $NaH_2PO_4$, 2 mM $CaCl_2$, 2 mM $MgSO_4$, 10 mM glucose, 26 mM $NaHCO_3$, 1 mM sodium pyruvate, and 5 µM L-glutathione

equilibrated with 95% $O_2$ and 5% $CO_2$, held at 35°C for 30–45 min, then maintained at room temperature.

## Recording

Individual brain slices were placed in a recording chamber where they were perfused at 4–5 ml/min with synthetic interstitial fluid (SIF) at 35°C that contained 126 mM NaCl, 3 mM KCl, 1.25 mM $NaH_2PO_4$, 1.6 mM $CaCl_2$, 1.5 mM $MgSO_4$, 10 mM glucose and 26 mM $NaHCO_3$ equilibrated with 95% $O_2$ and 5% $CO_2$. Somatic recordings were obtained under visual guidance (Axioskop FS2, Carl Zeiss, Oberkochen, Germany) using computer-controlled manipulators (Luigs and Neumann, Ratingen, Germany). Loose-seal cell-attached recordings were made using 3–5 MΩ impedance borosilicate glass pipettes (Warner Instruments, Hamden, CT, USA) that were filled with 140 mM NaCl, 23 mM glucose, 15 mM HEPES, 3 mM KCl, 1.5 mM $MgCl_2$, 1.6 mM $CaCl_2$ (pH 7.2 with NaOH, 300–310 mOsm/l). Whole-cell voltage clamp recordings were made using 3–5 MΩ pipettes filled with 135 mM $CsCH_3O_3S$, 10 mM QX-314, 10 mM HEPES, 5 mM phosphocreatine, 3.8 mM NaCl, 2 mM $Mg_{1.5}ATP$, 1 mM $MgCl_2$, 0.4 mM $Na_3GTP$, 0.1 mM $Na_4EGTA$ (pH 7.2 with CsOH, 290 mOsm/l). Whole-cell current clamp recordings were made using 10–15 MΩ pipettes filled with 130 mM $KCH_3SO_4$, 3.8 mM NaCl, 1 mM $MgCl_2$, 10 mM HEPES, 5 mM phosphocreatine di(tris) salt, 0.1 mM $Na_4EGTA$, 0.4 mM $Na_3GTP$, and 2 mM $Mg_{1.5}ATP$ (pH 7.3 with KOH; 290 mOsm/l). Electrophysiological records were acquired using a computer running Clampex 10 software (Molecular Devices, Palo Alto, CA, USA; RRID:SCR_011323) connected to a Multiclamp 700B amplifier (Molecular Devices) via a Digidata 1322 A digitizer (Molecular Devices). Data were low-pass filtered at 10 kHz and sampled at 50 kHz. Liquid junction potentials of 10 and 9 mV were accounted for in whole-cell voltage clamp and current clamp recordings respectively, and in voltage clamp recordings series resistance and membrane capacitance were corrected online. All recordings of autonomous action potential generation were made in the acute presence of 50 µM D-AP5, 20 µM DNQX, 10 µM GABAzine, and 2 µM CGP 55845 to block synaptic transmission.

## Two-photon imaging

MTS-roGFP-expressing neurons were imaged at 890 nm with 76 MHz pulse repetition and ~250 fs pulse duration at the sample plane. Two-photon excitation was provided by a G8 OPSL pumped Mira 900 F laser (Coherent, Santa Clara, CA, USA) and sample power was regulated by a Pockels cell electro-optic modulator (model M350-50-02-BK, Con Optics, Danbury, CT, USA). Images were acquired using an Ultima 2 P system running PrairieView 5 (Bruker Nano Fluorescence Microscopy, Middleton, WI, USA) and a BX51WI microscope (Olympus, Tokyo, Japan) with a 60 × 0.9 NA objective (UIS1 LUMPFL; Olympus). After baseline fluorescence had been measured, the maximum and minimum fluorescence were determined by the application of 2 mM dithiothreitol and then 200 µM aldrithiol-4 to fully reduce and oxidize the tissue, respectively. The relative oxidation at baseline, a measure of oxidative stress, was then calculated (*Sanchez-Padilla et al., 2014*).

## Immunohistochemistry and stereology

Mice were lightly anesthetized with isoflurane, deeply anesthetized with ketamine/xylazine (87/13 mg/kg i.p.) and then perfused transcardially with ~5 ml of phosphate buffered saline (PBS) followed by 30 ml of 4% formaldehyde in 0.1 M phosphate buffer (pH 7.4). Brains were removed and post-fixed for 2 hr in 4% formaldehyde, then washed in PBS. Brains were blocked and 70 µm thick coronal sections containing the STN were cut using a vibratome (VT1000S; Leica). Sections were washed in PBS and incubated for 48 hr at 4°C in anti-NeuN (clone A60; MilliporeSigma, Darmstadt, Germany; RRID:AB_2298772) at 1:200 in PBS with 0.2% Triton X-100 (MilliporeSigma) and 2% normal donkey serum. Sections were then washed in PBS and incubated for 90 min at room temperature in Alexa Fluor 488 donkey anti-mouse IgG (1:250; Jackson Immunoresearch, West Grove, PA, USA; RRID:AB_2340846) in 0.2% Triton X-100 and 2% normal donkey serum. Then the sections were washed in PBS and mounted on glass slides in Prolong Gold anti-fade medium (Thermo Fisher Scientific, Waltham, MA, USA).

NeuN labeled sections were imaged using an Axioskop two microscope (Carl Zeiss) with a 100 × 1.3 NA oil immersion objective (Plan-Neofluar 1018–595; Carl Zeiss). Unbiased stereological counting of STN neurons in a single hemisphere was performed using the optical fractionator technique

(*West et al., 1991*) as implemented in Stereo Investigator (MBF Bioscience, Williston, VT, USA; RRID:SCR_002526), using a counting frame of 50 µm × 50 µm × 8 µm and a grid size of 150 µm × 150 µm; all sections containing the STN were used for counting (~8 sections). STN volume was calculated from the sum of the areal extent of the STN on each section multiplied by the section thickness (70 µm). For all individual counts the Gundersen Coefficient of Error (CE) (*Gundersen et al., 1999*) was less than 0.1 (0.080 [0.075–0.090]), and the investigator performing the counting was blinded to the genotype of the mouse.

## Drugs

All drugs used in electrophysiology and imaging experiments were diluted to working concentration in SIF and bath applied. D-AP5, CGP 55845, DNQX, GABAzine (SR 95531), NMDA and gliclazide were purchased from Abcam (Cambridge, MA, USA). Glibenclamide, TFB-TBOA and DL-Dithiothreitol were purchased from Tocris Bioscience (Bristol, UK). Catalase (polyethylene glycol-catalase), aldrithiol-4 and MCS were purchased from Sigma-Aldrich (St. Louis, MO, USA).

## Data analysis and statistics

Electrophysiological data were analyzed using routines running in Igor Pro 6 and 7 (Wavemetrics, Portland, OR, USA; RRID:SCR_000325) or matplotlib (*Hunter, 2007*; RRID:SCR_008624). The firing rate of STN neurons was calculated from 1 min of recording or 100 action potentials (whichever covered the longer time period). Imaging data were analyzed using FIJI (*Schindelin et al., 2012*; RRID: SCR_002285). Statistical analyses were performed in Prism 5 (GraphPad Software, San Diego, CA, USA; RRID:SCR_002798) or R (http://www.r-project.org/; RRID:SCR_001905, RRID:SCR_000432). In order to make no assumptions about the distribution of the data, non-parametric statistical tests were used, and data are reported as *median [interquartile range]*; outliers were not excluded from the analysis. An $\alpha$-level of 0.05 was used for two-way statistical comparisons performed with the Mann-Whitney U test for unpaired data (represented with box plots), the Wilcoxon signed rank test for paired data (tilted line segment plots), Fisher's exact test for categorical data (bar plots) or the F-test for linear regression. Where datasets were used in multiple comparisons the p-value was adjusted to maintain the family-wise error rate at 0.05 using the Holm-Bonferroni method (*Holm, 1979*); adjusted p-values are denoted $p_h$. Box plots show median (central line), interquartile range (box) and 10–90% range (whiskers). For the primary findings reported in the manuscript, sample sizes for Mann-Whitney and Wilcoxon tests were estimated to achieve a minimum of 80% power using formulae described by *Noether (1987)*. The effect sizes used in these power calculations were estimated using data randomly drawn from uniform distributions (runif() function in R stats package). For Mann-Whitney tests, with a 50 percentile change in median between groups X and Y (the interquartile ranges of the groups don't overlap) $P(Y > X) \approx 0.88$ giving an estimation that at least 10 observations per group would be needed to achieve 80% power; for a 25 percentile change (the median of Y falls outside the interquartile range of X) $P(Y > X) \approx 0.72$ and the estimated requirement is at least 27 observations per group. For Wilcoxon tests, if all pairs of observations show the same direction of change, $P(X + X' > 0) = 1$ giving an estimation that at least 10 observations would be needed to achieve 80% power (note though that it is possible to show empirically that 6 observations gives 100% power in this case); if 90% of observations show the same direction of change, $P(X + X' > 0) \approx 0.98$ and the estimated requirement is at least 12 pairs of observations.

## Acknowledgements

This study was funded by CHDI Foundation and by NIH NINDS Grants 2 R37 NS041280 and 2 P50 NS047085. We thank Drs. Vahri Beaumont (CHDI) and Ignacio Munoz-Sanjuan (CHDI) for their comments on the work and Sasha Ulrich, Danielle Schowalter, and Marisha Alicea for management of mouse colonies.

## Additional information

### Funding

| Funder | Grant reference number | Author |
|---|---|---|
| CHDI Foundation | | Jeremy F Atherton<br>Mark D Bevan |
| National Institutes of Health | 2R37 NS041280 | Eileen L McIver<br>David L Wokosin<br>Mark D Bevan |
| National Institutes of Health | 2P50 NS047085 | Eileen L McIver<br>David L Wokosin<br>D James Surmeier<br>Mark D Bevan |

The funders had no role in study design, data collection and interpretation, or the decision to submit the work for publication.

### Author contributions

JFA, Designed the experiments, Conducted electrophysiology, imaging, immunohistochemistry and stereology experiments, Analyzed the data, Wrote the manuscript; ELM, Conducted electrophysiology, immunohistochemistry and stereology experiments; MRMM, Conducted immunohistochemistry and stereology experiments; DLW, Conducted imaging experiments; DJS, Designed experiments; MDB, Designed experiments, Conducted electrophysiology experiments, Wrote the manuscript, Directed the project

### Author ORCIDs

Mark D Bevan, http://orcid.org/0000-0001-9759-0163

### Ethics

Animal experimentation: This study was performed in accordance with the policies of the Society for Neuroscience and the National Institutes of Health. All animals were handled according to approved Institutional Animal Care and Use Committee protocols (IS00001185) of Northwestern University. All procedures were performed under isoflurane or ketamine/xylazine anesthesia, and every effort was made to minimize suffering.

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
