## [Decision Letter]

Thank you for submitting your article "Early dysfunction and progressive degeneration of the subthalamic nucleus in mouse models of Huntington's disease" for consideration by *eLife*. Your article has been favorably evaluated by Huda Zoghbi (Senior Editor) and three reviewers, one of whom is a member of our Board of Reviewing Editors. The following individual involved in review of your submission has agreed to reveal their identity: Margaret Rice (Reviewer #2).

The reviewers have discussed the reviews with one another and the Reviewing Editor has drafted this decision to help you prepare a revised submission.

Summary:

This is a very interesting study that implicates enhanced NMDAR signaling, leading to increased mitochondrial oxidant production, including that of hydrogen peroxide which triggers increased KATP activity, and results in slower and more variable autonomous STN neuronal firing in two HD mouse models (BACHD and Q175). This is one of the first studies to explore pathophysiological changes occurring early in the STN of HD mouse models. Overall the reviews are very positive about this work and recognize its important contribution to understanding the early pathogenesis in the STN using tow HD mouse models.

Essential revisions:

Revise the Abstract, Discussion and text of Results by removing conclusions on the role of glutamate uptake in the enhanced NMDA receptor-mediated oxidative stress.

*Reviewer #1:*

This body of work from Atherton et al. is a rather extensive study on the impact mutant Htt has on the STN. Involvement of the STN in HD has been debated over the years and this work, at least in mouse models of HD, provides the strongest evidence to data that the STN is a central aspect of HD pathology. A strength is that their observation/results were obtained using two full length Htt mouse models of HD. Starting at early stages of disease in these models they found that STN neurons had decreased intrinsic firing rate and related this to activation of KATP channels. This is followed by an elevated extra-synaptic NMDA activation by increased glutamate levels (results of decreased astrocyte uptake). Lastly, they show that at later stages, 12 months, there is a significant loss of STN neurons. Overall the data are strong making a convincible demonstration that the STN is involved in these models of HD. This is important as therapies for HD move to the clinic. This work provides a strong basis for the importance of assessing STN efficacy of treatments

*Reviewer #2:*

This report describes a thoughtful and thorough series of studies that not only illuminate a critical role for subthalamic nucleus (STN) neurons in mouse models of Huntington's disease, but also point to underlying mechanisms, including impaired glutamate uptake, NMDA receptor activation, H2O2 generation, and consequent K-ATP channel opening that slows STN neuron firing. This discussion carefully considers these factors in terms of cause versus effect. For all of these reasons, the studies should advance the field. There are a few generally minor concerns; most concern how the data are presented and discussed.

1) The notion that the direct pathway promotes "thoughts" is not obviously supported by the references cited (Introduction, first paragraph); cognitive function and thinking are not interchangeable terms.

2) The difference in frequency between WT and BACHD mice (is mean or median reported in the text?) was relatively small (7.9 vs 6.3 Hz, or a decrease of 20%); however, the examples in Figure 1 imply a four-fold difference. More accurately representative records should be used. The data shown in Figure 3 better represent the means reported; however, the much lower range for the untreated cells from BACHD mice (1.0 Hz) than those from BACHD reported in Figure 1 (6.3 Hz) needs some explanation. Were the untreated cells also recorded after the 3-5 h period as for AP5 pre-incubation?

3) The control data for MTS-roGFP imaging in the STN of BACHD mice appear to differ between Figure 3, with a lack of obvious difference between control BACHD in 3C and WT in 3B. Do statistical analyses indicate a difference between control BACHD data in 3C and WT, or between BACHD data in 3B and 3C? The meaning of "relative oxidation" as the y-axis title also needs clarification. How this was determined is explained in the Methods, but should also be noted briefly in the Results. As written, the text makes this murkier rather than clearer: "MTS-roGFP imaging revealed that the relative oxidant stress in BACHD STN neurons was elevated relative to WT."

4) Figure 6 shows data from the cell that had the greatest change with glicazide, not one representative of the mean (or median) response (6B). Similarly, 7B show a cell in NMDA with 3 spikes per 2 s, less than 2 Hz, whereas the lowest frequency in the range reported was 2.3 (subsection “NMDAR activation produces a persistent K_ATP_ channel-mediated disruption of autonomous activity in WT STN neurons”).

5) The results with exogenous NMDA do implicate NMDA-induced reactive oxygen species, but the more convincing experiment would have been to block NMDA receptors in BACHD mice (as in the MTS-roGFP imaging studies; Figure 4). They had an opportunity to test this in studies with Q175 mice (Figure 12), but although they show that glibenclamide and AP5 each lead to an increase in STN neurons firing rate in slices from Q175 mice, the key experiment of whether the effect of glibenclamide is absent when NMDA receptors are blocked by APV is not reported. This would strengthen the authors' conclusions, but is not essential.

6) It is not clear whether the decreased firing rate of STN neurons in slices from HD mice reflected inclusion of neurons that did not show autonomous activity (0 Hz). The frequency range for BACHD mice does not include 0 (subsection “The autonomous activity of STN neurons is disrupted in the BACHD model”), but the range for Q175 mice does (subsection “The STN of Q175 KI mice exhibits similar abnormalities to those observed in the BACHD model”).

H2O2Figuresthat *Reviewer #3:*

This is a very interesting study that implicates enhanced NMDAR signaling, leading to increased mitochondrial oxidative stress that triggers increased KATP activity, resulting in slowed, more variable autonomous STN neuronal firing in two HD mouse models (BACHD and Q175). This study includes a series of elegant experiments with results that convincingly demonstrate a mechanism for altered STN autonomous firing, implicating a pathological process independent of aberrant iSPN input to STN in HD.

One limitation, however, is that although the authors suggest this STN firing alteration underlies a variety of HD-related behaviors, this hypothesis is not tested. Moreover, they don't explore any in vivo recording to confirm altered STN firing or responsiveness to cortical and/or striatal iSPN input. As well, the link between these molecular alterations, aberrant autonomous firing, and STN neuronal loss is not established. The latter is complicated by the fact that the STN is part of the indirect pathway, which is impaired early at the level of striatal iSPNs (early loss of enkephalin and DARPP32, enhanced excitability, loss of LTP, increased extra-synaptic NMDAR signaling, etc.).

In addition, there are some specific concerns:

1) It seems counter-intuitive that by increasing the glutamate load (e.g., with pulse trains), a basal impairment in glutamate transport would no longer be observed; one would think it might instead be augmented. However, this could make sense in the context of a recent study by Armbruster et al. (2016), showing that train stimulation itself slows glutamate uptake, which might occlude a small difference in the NMDAR EPSC decay time constant, as the authors have postulated for the TFB-TBOA. The phenomenon reported by Armbruster et al. seems a more likely explanation for the difference in results, rather than what the authors suggest, that impaired glutamate uptake is restricted to the synapse. The latter seems unlikely since astrocytes ensheath more than one synapse.

2) Related to #1, the NMDAR component of the EPSC decay time constant can be quite variable (as shown in Figure 1; also see Vicini et al., 1998), and other factors (e.g. differences in access resistance, space clamp, etc.) could potentially impact the accuracy of measuring decay time in the context of such small currents. Moreover, the difference between the mean decay time constant in wild-type vs. BAC HD STN neurons is barely significant (P=0.0455). To further test the idea that impaired glutamate uptake is responsible for slowed decay for single NMDAR EPSCs, the authors could try using the glutamate scavenger GPT to see if this can accelerate decay time and show a greater effect in HD than wild-type, and/or add dextran to the extracellular solution to slow diffusion – this should magnify any differences in transporter function between the genotypes (see Mahfooz et al. NBD 2016 for approach).

3) In general, the focus on impaired glutamate uptake as a determinant of enhanced NMDAR-mediated oxidative stress seems unsupported. First, the authors rely heavily on previous studies showing reduced mRNA and/or protein expression levels of GLT1, but this does not prove impaired glutamate uptake under physiological conditions. In fact, few studies have measured clearance of synaptically released glutamate in brain slice: of these, one study in striatum demonstrated no impairment of glutamate uptake/clearance in R6/2 or YAC28 mice (Parsons et al., 2016), while two others show either minimal impairment with prolonged (200 ms), high concentration (3 mM) glutamate pulses (Dvorzhak et al., 2016) or slowing only of the late component of the iGluSnFr response (> 2 sec after synaptic release; Jiang et al., 2016).

Second, the difference in NMDAR EPSC decay shown in the current study is very small and is eliminated in the setting of trains of 5 stimuli at 50Hz (see points 1 and 2); since the authors mention that cortical input to STN frequently occurs in bursts, it seems unlikely that the Glu uptake impairment they see for single EPSCs could be an important contributor to increased oxidative stress in HD vs. wild-type STN. These points should be addressed in the Discussion, and less emphasis placed on the role of impaired glutamate uptake, which is not directly measured in this study.

4) The authors show increased mitochondrial oxidation and altered firing rate in BAC HD STN that is rescued by APV – this implicates NMDAR activity in these downstream events. However, it could be because of altered signaling via the NMDAR complex, i.e. due to NMDAR association with different effector proteins, rather than a result of increased calcium influx via NMDARs. This could be tested by comparing intracellular calcium transients in STN neurons between the genotypes using GCaMP6.

---

## [Author Response]

*Essential revisions:*

*Revise the Abstract, Discussion and text of Results by removing conclusions on the role of glutamate uptake in the enhanced NMDA receptor-mediated oxidative stress.*

As requested, we have revised the Abstract, Discussion and text of Results – removing the conclusions on the role of glutamate uptake in NMDA receptor-mediated oxidative stress and autonomous firing dysfunction.

*Reviewer #2:*

*[…] 1) The notion that the direct pathway promotes "thoughts" is not obviously supported by the references cited (Introduction, first paragraph); cognitive function and thinking are not interchangeable terms.*

The two sentences referring to the roles of the direct and indirect pathways have been changed to ‘The so-called direct pathway through the striatum promotes movement and ‘rewarding’ behavior through inhibition of GABAergic basal ganglia output (Chevalier and Deniau, 1990; Kravitz et al., 2010; Kravitz and Kreitzer, 2012). In contrast, the indirect pathway via the striatum, external globus pallidus and subthalamic nucleus (STN) and the hyperdirect pathway through the STN suppress the same processes through elevation of basal ganglia output (Maurice et al., 1999; Tachibana et al., 2008; Kravitz et al., 2010; Kravitz and Kreitzer, 2012).’

The reference Kravitz and Kreitzer, 2012, which deals with non-motor aspects of direct and indirect pathway encoding has been added.

*2) The difference in frequency between WT and BACHD mice (is mean or median reported in the text?) was relatively small (7.9 vs 6.3 Hz, or a decrease of 20%); however, the examples in Figure 1 imply a four-fold difference. More accurately representative records should be used. The data shown in Figure 3 better represent the means reported; however, the much lower range for the untreated cells from BACHD mice (1.0 Hz) than those from BACHD reported in Figure 1 (6.3 Hz) needs some explanation. Were the untreated cells also recorded after the 3-5 h period as for AP5 pre-incubation?*

All data in the manuscript are reported in the form *median [interquartile range]*. We agree that the BACHD example cells shown in Figure 1 do not exemplify the difference in median frequency between WT and BACHD mice. The neurons from BACHD mice comprise a phenotypic population that displays compromised autonomous firing, and neurons that have relatively normal firing. We have therefore amended Figure 1 with the addition of a histogram illustrating the distribution of autonomous firing for WT and BACHD STN neurons. As can be seen from this histogram, a BACHD neuron that illustrated the median firing rate, would not be representative of either firing mode. We therefore chose to illustrate a neuron that is representative of phenotypic neurons. In addition to amending Figure 1, we have amended the text in the Results section and the legend for Figure 1to clarify this point. The factors that could underlie the disrupted activity of a subset of neurons in the BACHD model are discussed.

The strength of the BACHD firing phenotype varies across animals and the difference between the data in Figure 3 and that in Figure 1 is explained by the smaller sample in Figure 3 from this variable population. In these D-AP5 pre-treatment experiments, recordings from treated and untreated slices were interleaved. Because an untreated slice was usually examined first, observations were on average 1–2 hours earlier than for treated slices. This is not expected to influence the result because in the full BACHD dataset there was no correlation between time ex vivo and firing frequency (r^2^ = 0.0054, p = 0.4151, n = 126).

*3) The control data for MTS-roGFP imaging in the STN of BACHD mice appear to differ between Figure 3, with a lack of obvious difference between control BACHD in 3C and WT in 3B. Do statistical analyses indicate a difference between control BACHD data in 3C and WT, or between BACHD data in 3B and 3C? The meaning of "relative oxidation" as the y-axis title also needs clarification. How this was determined is explained in the Methods, but should also be noted briefly in the Results. As written, the text makes this murkier rather than clearer: "MTS-roGFP imaging revealed that the relative oxidant stress in BACHD STN neurons was elevated relative to WT."*

(Note, this comment appears to be in reference to the data in Figure 4 not Figure 3)

The WT data shown in Figure 4 and the control BACHD data shown in Figure 4 are not significantly different (p = 0.1666, Mann-Whitney U test). There is also no significant difference between the BACHD data in Figure 4 and the untreated BACHD data in Figure 4 (p = 0.2089, Mann-Whitney U test). However, these comparisons are not valid because the data in 4Bwas acquired 15 months prior to the data in 4C. Although we strive to maintain identical experimental conditions, experiments separated by 15 months likely utilized different batches of virus and less than identical imaging conditions. In hindsight, it would have been optimal to collect all the data in Figure 4 at the same time, particularly if our intention was to compare the absolute levels of relative oxidation between all the experimental groups. However, the experiment in 4Bwas designed to test whether the relative oxidant stress of mitochondria was greater in BACHD vs. WT STN neurons and the experiment in 4C (15 months later) was designed to test whether NMDAR antagonism could lower the relative oxidant stress of mitochondria in BACHD STN neurons. These comparisons were confirmed statistically and the conclusions from these experiments are also supported by the experiments in the rest of the paper.

Precisely what the relative oxidation measurement represents has been clarified further in the Results section.

*4) Figure 6 shows data from the cell that had the greatest change with glicazide, not one representative of the mean (or median) response (6B). Similarly, 7B show a cell in NMDA with 3 spikes per 2 s, less than 2 Hz, whereas the lowest frequency in the range reported was 2.3 (subsection “NMDAR activation produces a persistent KATP channel-mediated disruption of autonomous activity in WT STN neurons”).*

We have changed the example cell used in Figure 6, and selected a more representative segment of firing for the control trace in Figure 7.

*5) The results with exogenous NMDA do implicate NMDA-induced reactive oxygen species, but the more convincing experiment would have been to block NMDA receptors in BACHD mice (as in the MTS-roGFP imaging studies; Figure 4). They had an opportunity to test this in studies with Q175 mice (Figure 12), but although they show that glibenclamide and AP5 each lead to an increase in STN neurons firing rate in slices from Q175 mice, the key experiment of whether the effect of glibenclamide is absent when NMDA receptors are blocked by APV is not reported. This would strengthen the authors' conclusions, but is not essential.*

We have now performed this experiment and added the following text to the Results section: ‘As described above, 3–5-hour NMDAR antagonism with D-AP5 partially rescued autonomous activity in BACHD STN neurons. […] Thus, these observations are consistent with the conclusion that prolonged NMDAR antagonism partially rescued autonomous activity in BACHD STN neurons through a reduction in K_ATP_ channel-mediated firing disruption.’

*6) It is not clear whether the decreased firing rate of STN neurons in slices from HD mice reflected inclusion of neurons that did not show autonomous activity (0 Hz). The frequency range for BACHD mice does not include 0 (subsection “The autonomous activity of STN neurons is disrupted in the BACHD model”), but the range for Q175 mice does (subsection “The STN of Q175 KI mice exhibits similar abnormalities to those observed in the BACHD model”).*

Neurons that did not show autonomous activity were included in the firing rate analyses but could not be included in the CV analyses for obvious reasons, as reflected in the lower n for CV data versus frequency data. The ranges described here represent the interquartile range and not the minimum to maximum range. The fact that the interquartile range in BACHD mice does not include silent neurons, whereas the range in Q175 mice does, is a reflection of the relatively severe disruption of autonomous firing in Q175 mice.

*Reviewer #3:*

*[…] One limitation, however, is that although the authors suggest this STN firing alteration underlies a variety of HD-related behaviors, this hypothesis is not tested. Moreover, they don't explore any in vivo recording to confirm altered STN firing or responsiveness to cortical and/or striatal iSPN input. As well, the link between these molecular alterations, aberrant autonomous firing, and STN neuronal loss is not established. The latter is complicated by the fact that the STN is part of the indirect pathway, which is impaired early at the level of striatal iSPNs (early loss of enkephalin and DARPP32, enhanced excitability, loss of LTP, increased extra-synaptic NMDAR signaling, etc.).*

The similarity of HD symptoms to those arising from STN lesion or inactivation (Crossman et al., 1988; Hamada and DeLong, 1992; Baunez and Robbins, 1997; Bickel et al., 2010; Jahanshahi et al., 2015), combined with evidence of abnormal STN activity in HD models in vivo (Callahan and Abercrombie, 2015a, b) and STN neuronal degeneration in HD (Lange et al., 1976; Guo et al., 2012), led to our hypothesis that STN neurons are dysfunctional in HD. Now that we have confirmed that STN neurons are indeed dysfunctional in two HD models, we can next pursue whether STN dysfunction contributes to abnormal behavior in these HD models. We are currently evaluating several approaches based on the novel findings presented here that might ameliorate STN dysfunction in HD mice in vivo. However, these experiments will likely generate another full manuscript and are therefore beyond the scope of the current report. We concur that we have not yet established whether the STN dysfunction and neurodegeneration that we have described for the first time contribute to abnormal behavior in these models and have tempered the manuscript accordingly. We also agree that we have not yet established that the neuronal loss that we observed is directly related to the increased NMDAR-dependent, oxidative stress of STN neurons, although this is a well-recognized cell death pathway. We are therefore currently testing whether correcting or compensating for STN dysfunction, using strategies suggested by the abnormalities reported here, will reduce or eliminate cell death.

*In addition, there are some specific concerns:*

*1) It seems counter-intuitive that by increasing the glutamate load (e.g., with pulse trains), a basal impairment in glutamate transport would no longer be observed; one would think it might instead be augmented. However, this could make sense in the context of a recent study by Armbruster et al. (2016), showing that train stimulation itself slows glutamate uptake, which might occlude a small difference in the NMDAR EPSC decay time constant, as the authors have postulated for the TFB-TBOA. The phenomenon reported by Armbruster et al. seems a more likely explanation for the difference in results, rather than what the authors suggest, that impaired glutamate uptake is restricted to the synapse. The latter seems unlikely since astrocytes ensheath more than one synapse.*

We thank the reviewer for bringing to our attention this potentially relevant finding. As this paper was published after our initial submission, we were unable discuss it in the original manuscript, but have now added discussion of this paper to the revised manuscript.

*2) Related to #1, the NMDAR component of the EPSC decay time constant can be quite variable (as shown in Figure 1; also see Vicini et al., 1998), and other factors (e.g. differences in access resistance, space clamp, etc.) could potentially impact the accuracy of measuring decay time in the context of such small currents. Moreover, the difference between the mean decay time constant in wild-type vs. BAC HD STN neurons is barely significant (P=0.0455). To further test the idea that impaired glutamate uptake is responsible for slowed decay for single NMDAR EPSCs, the authors could try using the glutamate scavenger GPT to see if this can accelerate decay time and show a greater effect in HD than wild-type, and/or add dextran to the extracellular solution to slow diffusion – this should magnify any differences in transporter function between the genotypes (see Mahfooz et al. NBD 2016 for approach).*

We have compared the access resistances associated with the NMDAR current measurements in this study. They were not significantly different across each experimental group (p = 0.2297), and the decay time constant was not correlated with access resistance (r^2^ = 0.0838, p = 0.18). Technical reasons are therefore unlikely to have produced a consistent measurement error.

The suggestions for further experiments and the iGluSNr approaches described below will be certainly be applied in subsequent studies as we further dissect the mechanisms underlying the significantly extended NMDAR currents in STN neurons in HD models.

*3) In general, the focus on impaired glutamate uptake as a determinant of enhanced NMDAR-mediated oxidative stress seems unsupported. First, the authors rely heavily on previous studies showing reduced mRNA and/or protein expression levels of GLT1, but this does not prove impaired glutamate uptake under physiological conditions. In fact, few studies have measured clearance of synaptically released glutamate in brain slice: of these, one study in striatum demonstrated no impairment of glutamate uptake/clearance in R6/2 or YAC28 mice (Parsons et al., 2016), while two others show either minimal impairment with prolonged (200 ms), high concentration (3 mM) glutamate pulses (Dvorzhak et al., 2016) or slowing only of the late component of the iGluSnFr response (> 2 sec after synaptic release; Jiang et al., 2016).*

*Second, the difference in NMDAR EPSC decay shown in the current study is very small and is eliminated in the setting of trains of 5 stimuli at 50Hz (see points 1 and 2); since the authors mention that cortical input to STN frequently occurs in bursts, it seems unlikely that the Glu uptake impairment they see for single EPSCs could be an important contributor to increased oxidative stress in HD vs. wild-type STN. These points should be addressed in the Discussion, and less emphasis placed on the role of impaired glutamate uptake, which is not directly measured in this study.*

We accept these critiques of our assumption that glutamate uptake impairment accounts for NMDAR-mediated oxidative stress and firing disruption made in both points 2 & 3 above. We have therefore removed this conclusion from the manuscript and will make a more in-depth investigation of this potential linkage in future studies.

*4) The authors show increased mitochondrial oxidation and altered firing rate in BAC HD STN that is rescued by APV – this implicates NMDAR activity in these downstream events. However, it could be because of altered signaling via the NMDAR complex, i.e. due to NMDAR association with different effector proteins, rather than a result of increased calcium influx via NMDARs. This could be tested by comparing intracellular calcium transients in STN neurons between the genotypes using GCaMP6.*

Again, we agree with the reviewer that altered NMDAR and associated signaling pathway properties could contribute to NMDAR-mediated oxidative stress and firing disruption and have therefore modified our interpretation and discussion of the data accordingly throughout the manuscript.